# Cholinergic-like neurons carrying PSEN1 E280A mutation from familial Alzheimer's disease reveal intraneuronal sAPPβ fragments accumulation, hyperphosphorylation of TAU, oxidative stress, apoptosis and Ca$^{2+}$ dysregulation: Therapeutic implications

**Viviana Soto-Mercado, Miguel Mendivil-Perez, Carlos Velez-Pardo, Francisco Lopera, Marlene Jimenez-Del-Rio** *

Neuroscience Research Group, Medical Research Institute, Faculty of Medicine, University of Antioquia (UdeA), SIU Medellin, Medellin, Colombia

* marlene.jimenez@udea.edu.co

## Abstract

Alzheimer's disease (AD) is a neurodegenerative disorder characterized by progressive memory loss and cognitive disturbance as a consequence of the loss of cholinergic neurons in the brain, neuritic plaques and hyperphosphorylation of TAU protein. Although the underlying mechanisms leading to these events are unclear, mutations in presenilin 1 (PSEN1), e.g., E280A (PSEN1 E280A), are causative factors for autosomal dominant early-onset familial AD (FAD). Despite advances in the understanding of the physiopathology of AD, there are no efficient therapies to date. Limitations in culturing brain-derived live neurons might explain the limited effectiveness of AD research. Here, we show that mesenchymal stromal (stem) cells (MSCs) can be used to model FAD, providing novel opportunities to study cellular mechanisms and to establish therapeutic strategies. Indeed, we cultured MSCs with the FAD mutation PSEN1 E280A and wild-type (WT) PSEN1 from umbilical cords and characterized the transdifferentiation of these cells into cholinergic-like neurons (ChLNs). PSEN1 E280A ChLNs but not WT PSEN1 ChLNs exhibited increased intracellular soluble amyloid precursor protein (sAPPf) fragments and extracellular Aβ42 peptide and TAU phosphorylation (at residues Ser202/Thr205), recapitulating the molecular pathogenesis of FAD caused by mutant PSEN1. Furthermore, PSEN1 E280A ChLNs presented oxidative stress (OS) as evidenced by the oxidation of DJ-1Cys$^{106}$-SH into DJ-1Cys$^{106}$-SO$_3$ and the detection of DCF-positive cells and apoptosis markers such as activated pro-apoptosis proteins p53, c-JUN, PUMA and CASPASE-3 and the concomitant loss of the mitochondrial membrane potential and DNA fragmentation. Additionally, mutant ChLNs displayed Ca$^{2+}$ flux dysregulation and deficient acetylcholinesterase (AChE) activity compared to control ChLNs. Interestingly, the inhibitor JNK SP600125 almost completely blocked TAU phosphorylation. Our findings demonstrate that FAD MSC-derived cholinergic neurons with the

**Data Availability Statement:** All relevant data are within the manuscript and its Supporting Information files.

**Funding:** This study was funded by MinCiencias grant # 1115-807-62912, contract #749-2018 to MJ-Del-Rio, CV-P and MM-P. VS-M is a doctoral student of the Basic Biomedical Sciences Academic Corporation program at the Universidad de Antioquia (UdeA), and is funded by the 2019 Bicentennial Doctoral Excellence Scholarship, MinCiencias-Colombia.

**Competing interests:** The authors have declared that no competing interests exist.

PSEN1 E280A mutation provide important clues for the identification of targetable pathological molecules.

## Introduction

Alzheimer's disease (AD) is a chronic neurodegenerative condition characterized by loss of memory, reasoning and decision-making functions [1] due to the severe loss of cholinergic neurons from the nucleus basalis magnocellularis of Meynert and cholinergic projections to the cortex and hippocampus [2]. The neuropathological profile of AD is associated with the extracellular accumulation of insoluble forms of amyloid-β (Aβ) in plaques and intracellular aggregation of the microtubule protein TAU in neurofibrillary tangles [3]. Aβ is derived by the proteolytic cleavage of amyloid β precursor protein (APP). The full length APP is first cleaved by β-secretase (BACE1) freeing a soluble proteolytic fragment, recognized as soluble APPβ (sAPPβ). The remaining C-terminal membrane bound APP fragment, CTFβ or C99 fragment, undergoes additional cleavages by γ-secretase to generate a series of peptides prone to aggregation [4]. Interestingly, both hAPP and Aβ species have been shown to aggregate intracellularly [5–7]. Most mutations in the presenilin 1 (*PSEN 1)* gene, which codes for the catalytic component of γ-secretase [8], result in the overproduction of Aβ, specifically, the 42-amino acid Aβ isoform (Aβ$_{1-42}$, hereafter Aβ$_{42}$) [9], and occur most frequently in familial AD (FAD; http://www.molgen.ua.ac.be/ADMutations/). Glu280Ala (p. E280A, c.839A > C, exon 8) in *PSEN1* is a well-characterized FAD mutation found in a large kindred localized in Antioquia, Colombia [10–13] that shows typical phenotypes of AD with complete penetrance [14]. Similar to the majority of dominant-negative *PSEN1* mutations [15, 16], PSEN1 E280A produces increased Aβ$_{42}$ deposition [17], hippocampal neuron loss [18], and Aβ/TAU accumulation in young adults [19, 20].

Despite advances in the understanding of the physiopathology of AD [21], there are no efficient therapies to date. Although limitations in culturing brain-derived live neurons might slow AD research, the rapid advances in cellular genetic reprogramming, in particular the induction of somatic cells (e.g., fibroblast) into stem cells (e.g., human induced pluripotent stem cells, hiPSCs), has led to the modeling of FAD PSEN1 mutations *in vitro* [22–25]. Obtaining iPSCs from patients bearing *PSEN1* mutations is appealing; however, the isolation and purification procedures are technically challenging, expensive, time consuming and labor intensive. Alternatively, the human mesenchymal stromal (stem) cells derived from Wharton's jelly tissue (WJ-MSCs) are multipotent cells that can differentiate and/or transdifferentiate into mesodermal and ectodermal lineage cells [26–29]. Because MSCs might be equivalent to human embryonic stem cells (hESCs) and hiPSCs [30, 31]; these cells have become an interesting and promising tool for modeling FAD PSEN1 E280A *in vitro*.

The aim of the present study was to establish an *in vitro* cellular model that reveals the major pathologic features of the FAD PSEN1 E280A mutation, thereby enabling investigation of the pathomechanisms of early onset FAD. Therefore, Aβ42 accumulation, Aβ42 production, TAU phosphorylation, oxidative stress (OS), cell death, and neuronal dysfunction were investigated in cholinergic-like neurons (ChLNs) derived from wild-type (control) and PSEN1 E280A MSCs. We demonstrate for the first time that FAD PSEN1 E280A pathology can be recapitulated in MSC-derived ChLNs. These findings in ChLNs show great promise for modeling human FAD *in vitro* and identifying therapeutic targets for AD treatment.

## Materials and methods

The collection and use of umbilical cords from newborns was approved by Ethics Committee of the Hospital San Vicente Fundacion Research act # 13–2015 Colombia, and was provided following natural childbirth with written consent. Donors had a familial background of AD. The mother's medical history was negative for human pathogens, such as human immunodeficiency virus 1/2, hepatitis B and C virus, and syphilis. The cord (~7 cm long) was immersed in low-glucose DMEM (Sigma) supplemented with 100 U Penicillin/streptomycin (Sigma) and 5 µg/ml Plasmocin (Invivogen) and immediately transported to the laboratory.

### Isolation and expansion of hWJ-MSCs

The human umbilical cords were obtained from ten healthy, natural childbirths (Tissue Bank Code (TBC) # WJMSC-11, -12, -13, -14, -15, -16, -17, -18, -19, -20) and aseptically stored at 4 ˚C PBS containing 1% penicillin and streptomycin. The cords were rinsed several times to drain blood from vessels, cut into 2–3-cm-long segments and rinsed again. Umbilical arteries and veins were removed, and the remaining tissue was transferred to a sterile container and chopped into small fragments in PBS. The explants were digested with an enzyme mixture containing 0.25% trypsin, 0.1% Dispase and 0.5% collagenase II for 2 h at 37 ˚C under constant agitation. Then, the digestion products were centrifuged at 447 x $g$ for 40 min, and the pellet was cultured in T75 cell culture flasks (Corning) in hWJ-MSC regular culture medium (low-glucose DMEM supplemented with 20% fetal bovine serum (FBS, Sigma), 100 U penicillin/ streptomycin and 5 µg/ml Plasmocin). Once confluence had been reached, adherent cells (passage 0) were detached with 0.25% trypsin and passaged at 13,000 cells/ cm$^2$ in a T75 flask. Cells from passages 2 or 4 were harvested during the first expansion period for further characterization and cryopreservation.

### Identification of the PSEN1 E280A mutation in WJMSCs

The PSEN1 E280A mutation was detected by PCR using mismatch primers and digestion of the products with *Bsm1* [32]. Digested products were separated on a 3% agarose gel. According to different mobility electrophoretic patterns, samples were classified as wild-type (WT) or mutant PSEN1 E280A when compared to PSEN1 E280A carrier (positive case NeuroBank (NB) code #18233) or wild type PSEN1 genotype (NB code#18574). The TBC# WJMSC-12 (female) sample was identified for PSEN1 E280A mutation. For comparative purposes, we pre-selected female WT PSEN1 samples (TBC# WJMSC-11, -15, -17), and TBC# WJMSC-11 WT PSEN1 was randomly selected for further experiments.

**APOE genotyping analysis.** Genotyping of the APOE polymorphism (n = 10) was performed using polymerase chain reaction amplification of a 244-bp fragment followed by digestion with *HhaI* as described by [33].

**Karyotyping.** Karyotype analysis was performed by the Medical Genetics Unit of the Faculty of Medicine–UdeA, using standard cytogenetic protocols. At 60–70% confluence, WJMSC-11 WT and WJMSC-12 PSEN1 E280A cells were incubated with 0.1 mg/ml Colcemid (Sigma) for 90 min at 37 ˚C. Then, the cells were detached with 0.25% trypsin and centrifuged at 591 x $g$ for 20 min. The medium was removed, and the hypotonic solution (0.075 M KCl, 0.017 M Na-citrate) was added and incubated for 20 min at 37 ˚C. After a new centrifugation, cells were fixed with freshly prepared Carnoy's solution. Metaphase spreads were analyzed after staining with quinacrine (Sigma) for karyotyping. Analysis was performed on three different primary cultures counting 20 metaphases for each sample.

## Colony-forming units assay

The colony formation assay is an *in vitro* cell survival assay based on the ability of a single cell to grow into a colony [34]. WJMSC-11 WT and WJMSC-12 PSEN1 E280A cells were seeded at a density of 200 cells/ well on 6-well plates followed by the addition of 3 mL of regular culture medium. The cultures were left to grow in a humidified atmosphere with 5% $CO_2$ at 37 ˚C for 15 days. The culture medium was changed twice a week. After 15 days of cultivation, both WT and mutant PSEN-1 cells were stained with 0.5% crystal violet and counted using the cell counter plugin from ImageJ program. The experiment was conducted three times.

## Immuno-phenotypic characterization

Standard flow cytometry techniques were used to determine the cell surface epitope profile (CD9, CD73, CD90, CD34 and CD45) of both WT and mutant PSEN1 MSCs. Briefly, hWJ-MSCs were incubated with saturating concentrations (1:500) of mouse monoclonal antibodies conjugated to human CD9-peridinin chlorophyll protein (PerCP)-cy5.5, CD73-phycoerythrin (PE), CD90 PE-cy5.5, CD34 PE, and CD45- fluorescein isothiocyanate (FITC). All antibodies were purchased from BD Biosciences (San Diego, CA). Cells were incubated for 1 h at 4 ˚C. Prior to antibody labeling, the cells were preincubated with 5% fetal bovine sera (FBS) for 10 min to block nonspecific binding. Cell suspensions were washed and resuspended in PBS for analysis on an LSRFortessa (BD Biosciences). Ten thousand events were acquired, and the acquisition analysis was performed using FlowJo 7.6.2. Positive staining was defined as the fluorescence emission that exceeded levels obtained by more than 99% of cells from the population stained with the corresponding negative controls. The isotype (negative) control used in this study was IgG1 PE-Cy5.5, IgG1-PE and IgG1- FITC (BD Biosciences).

## Cell differentiation

**Adipogenic differentiation.** Adipogenic differentiation was performed according to [35] with minor modifications. Briefly, WT and mutant MSCs at passages 4–7 were plated at a density 20,000 cells/$cm^2$ in a 12-well plate in regular culture medium. At 90–100% confluence, the culture medium was replaced by adipogenic induction medium, including high-glucose DMEM, 10% FBS, 0.5 mM 3-isobutyl-1-methylxanthine (Sigma, cat # I5879), 100 μM indomethacin (Sigma, cat # I7378), 0.1 μM dexamethasone and 10 μg/ mL insulin. After 28 days, cells were fixed with 4% formaldehyde (FA) and immediately stained with the standard Oil-Red-O protocol. Control cells were kept in regular culture medium.

**Osteogenic differentiation.** Osteogenic differentiation was performed according to [35] with minor modifications. Briefly, WT and mutant cells at passages 4–7 were plated at a density of 10,000 cells/ $cm^2$ in 12-well plates in regular culture medium. After 72 h, the culture medium was replaced by osteogenic differentiation medium containing high-glucose DMEM (Sigma), 10% FBS, 1 μM dexamethasone (Alfa Aesar, cat # A17590), 250 μM sodium ascorbate (Sigma, cat # A4034), and 10 mM β-glycerophosphate (Alfa Aesar, cat # L03425). The medium was changed every 3–4 days. Control cells were kept in regular culture medium. After 28 days of induction, cells were fixed in 4% FA and stained with standard Von Kossa Staining.

**Chondrogenic differentiation.** Chondrogenic differentiation was performed according to ref. [36] with minor modifications. Briefly, $2.5 \times 10^5$ WT and mutant cells were left aggregated in microwell plates and then provided with chondrogenic medium containing high-glucose DMEM, 10% FBS, 10 μg/ L TGF-β3, 0.1 μmol/ L dexamethasone, 50 μmol/ L vitamin C, and 6.25 mg/L insulin. The medium was changed every 3–4 days. Control cells were kept in regular culture medium. After 28 days of induction, cells were fixed in 4% formaldehyde, stained with toluidine blue for 2 min at room temperature and viewed by light microscopy.

**Cholinergic-Like Neuron (ChLN) differentiation.** ChLN differentiation was performed according to ref. [29]. WT and mutant MSCs were seeded at $1–1.5 \times 10^4$ cells/ cm$^2$ in laminin-treated culture plates for 24 h in regular culture medium (RCm). Then, the medium was removed, and cells were incubated in minimal culture medium (hereafter MCm) containing low-glucose DMEM and 1% FBS or in cholinergic differentiation medium (*Cholinergic-N-Run medium*, hereafter Ch-N-Rm) containing DMEM/F-12 media 1:1 Nutrient Mixture (Gibco cat# 10565018), 10 ng/ mL basic fibroblast growth factor (bFGF) recombinant human protein (Gibco Cat# 13256029), 50 μg/ mL sodium heparin (Hep, Sigma-Aldrich cat# H3393), 0.5 μM all-trans retinoic acid, 50 ng/ml sonic hedgehog peptide (SHH, Sigma cat# SRP3156) and 1% FBS at 37 ˚C for 7 days. After this process of transdifferentiation, the cells were labeled as WT PSEN1 or PSEN1 E280A ChLNs. Since Ch-N-Rm contains several factors that might interfere with the experiment interpretation and measurements, WT PSEN1 and PSEN1 E280A ChLNs (obtained after 7 days in Ch-N-Rm) were left in regular culture medium (RCm) for 0, 2 or 4 additional days, hereafter 0, 2, or 4 days of post transdifferentiation.

## Immunofluorescence analysis

For the analysis of neural-, Alzheimer's disease-, oxidative stress- and cell death-related markers, the cells treated under different conditions were fixed with cold ethanol (-20 ˚C) for 20 min, followed by Triton X-100 (0.1%) permeabilization and 10% bovine serum albumin (BSA) blockage. Cells were incubated overnight with primary neural antibodies against glial fibrillary acidic protein (GFAP 1:200, cat# sc6170, Santa Cruz), microtubule-associated protein 2 (MAP2, 1:250, cat MA1-25044, Invitrogen), β-tubulin III (1:250, cat# G712 A, Promega) and choline-acetyltransferase (ChAT, 1:50, cat# AB144 P, Millipore); primary antibodies against APP$_{751}$ and/or protein amyloid β$_{1–42}$ (1:500; clone 6E10 cat# 803014, Biolegend), total TAU (1: 500; t-Tau; cat# T6402, Sigma), and phospho-TAU (p-Tau, 1:500, Ser202/Thr205, cat# MN1020 (AT8), Thermo Fisher Scientific); and primary antibodies against oxidized DJ-1 (1:500; ox(Cys106)DJ1; spanning residue C$^{106}$ of human PARK7/DJ1; oxidized to produce cysteine sulfonic (SO$_3$) acid; cat # MABN1773, Millipore). To assess cell death, we used primary antibodies against p53-upregulated modulator of apoptosis (1:500; PUMA, cat# ab-9643, Abcam), p53 (1:500; cat# MA5-12453, Millipore), phospho-c-Jun (1:250; c-Jun (S63/73) cat# sc-16312, Santa Cruz), and caspase-3 (1:250; cat # AB3623, Millipore). After exhaustive rinsing, we incubated the cells with secondary fluorescent antibodies (DyLight 488 and 594 horse anti-rabbit, -goat and -mouse, cat DI 1094, DI 3088, and DI 2488, respectively) at 1:500. The nuclei were stained with 1 μM Hoechst 33342 (Life Technologies), and images were acquired on a Floyd Cells Imaging Station microscope.

## Western blot analysis

Cells were incubated as described above, detached with 0.25% trypsin and lysed in 50 mM Tris-HCl, pH 8.0, with 150 mM sodium chloride, 1.0% Igepal CA-630 (NP-40), and 0.1% sodium dodecyl sulfate and a protease inhibitor cocktail (Sigma-Aldrich). All lysates were quantified using the bicinchoninic acid assay (Thermo Scientific cat # 23225). Extracted samples (30 μg of proteins) were heated at 95 ˚C for 5 min in 2 x SDS and 20x reducing agent (except for protein oxDJ-1) and loaded into 12% Bis/Tris gels at 120 V for 90 min, and the bands were transferred onto nitrocellulose membranes (Hybond-ECL, Amersham Biosciences) at 270 mA for 90 min using an electrophoretic transfer system (BIO-RAD) according to ref. [37] with minor modifications. The membranes were incubated overnight at 4 ˚C with anti-GFAP, MAP2, β-tubulin III, ChAT, amyloid β$_{1–42}$, total TAU, phospho-TAU, ox(Cys$^{106}$) DJ1, PUMA, p53, p-c-Jun, and caspase-3 primary antibodies (1:5000). The anti-actin antibody

(1:1000, cat #MAB1501, Millipore) was used as an expression control. Secondary infrared antibodies (goat anti-rabbit IRDye® 680RD, cat #926–68071; donkey anti-goat IRDye ® 680RD, cat # 926–68074; and goat anti mouse IRDye ® 800CW, cat #926–32270; LI-CORBiosciences) at 1:1000 were used for western blotting analysis, and data were acquired using Odyssey software. To directly control the conformation-dependent differences among Aβ assemblies, we prepared a homogenous synthetic unaggregated (i.e., monomers) and large oligomeric Aβ$_{42}$ assemblies according to ref. [38]. Briefly, after solubilization of the peptide (Sigma Cat #A9810) in DMSO, the "unaggregated" peptide was obtained by dissolving the DMSO-solubilized peptide in water and used immediately (0 days). To obtain the "large oligomers", 10 mM Tris was added to DMSO-solubilized peptide solution and incubated it for 15 days at 4 ˚C. The determination of the aggregation state of Aβ$_{42}$ was performed by Western analysis of SDS-PAGE as described above. The assessment was repeated three times in independent experiments.

## Mass spectrometry analysis

Gel region between 60 and 125 kDa was excised from the experimental lane, cut into 1 x 1 mm cubes, incubated consequently with 10 mM dithiothreitol and 55 mM iodacetamid in 100 mM ammonium bicarbonate, and dehydrated with acetonitrile. Then, the proteins were in-gel digested overnight at 37 ˚C with LysC protease (Promega, Mannheim). The resulting peptide mixture was extracted twice with exchange of 5% formic acid and 100% acetonitrile, the extracts pulled together and dried down. The peptides were re-suspended in 25 μl of 5% formic acid and 5 μl were taken for LC-MS/MS analysis. The analysis was performed on a nano-UPLC Ultimate 3000 interfaced on-line to a Q-Exactive HF Hybrid Quadrupole-Orbitrap mass spectrometer (both Thermo Fischer Scientific, Bremen). The UPLC system was equipped with Acclam PepMap™ 100 75 μm x 2 cm trapping column and 75 μm x 15 cm separating column packed with 3 μm diameter C18 particles (Thermo Fischer Scientific, Bremen). Peptides were separated using 180 min linear gradient (solvent A– 0.1% formic acid in water, solvent B– 0.1% formic acid in neat acetonitrile). Spectra were acquired in DDA mode using top 20 method. Spectra were then converted into (.mgf) format and searched human sequences in the UniProt database (January 2020) using MASCOT software (version 2.2.04) and against human Amyloid-beta precursor protein (ACC No P05067) without enzyme specificity. Mass tolerance was set on 5 ppm and 0.025 Da for precursor and fragment ions respectively; variable modification–Cysteine carbamidomethyl and propionamide, methionine oxidation, protein N-terminal acetyl. The result of the database search was evaluated by Scaffold software (v.4.7.5, Proteome Software, Portland) and the spectra matched Amyloid-beta precursor protein also manually inspected. To investigate the presence of Aβ$_{42}$, the spectra were acquitted in DDA targeted mode using masses of C- and N-terminal peptides which are unique for 671–713 fragment, and then processed by SkyLine software.

## Analysis of cells

**Evaluation of intracellular hydrogen peroxide (H$_2$O$_2$) by fluorescence microscopy.** To determine the levels of intracellular H$_2$O$_2$, we used 2′,7′-dichlorofluorescein diacetate (5 μM, DCFH2-DA; Invitrogen). hWJ-MSCs or ChLNs were left in RCm for 0, 2 and 4 days. Then, the cells (5 x 10$^3$) were incubated with the DCFH$_2$-DA reagent for 30 min at 37 ˚C in the dark. Cells were then washed, and DCF fluorescence intensity was determined by analysis of fluorescence microscopy images. The assessment was repeated three times in independent experiments. The nuclei were stained with 0.5 μM Hoechst 33342 (2.5 μM) staining compound. The assessment was repeated three times in independent experiments blind to experimenter.

**Evaluation of intracellular hydrogen peroxide ($H_2O_2$) by flow cytometry.** $H_2O_2$ was determined with 2',7'-dichlorofluorescein diacetate (1 μM, DCFH$_2$-DA). ChLNs were left in regular medium (RCm) for 0, 2 and 4 days. Then, the cells ($1\times10^5$) were incubated with DCFH$_2$-DA reagent for 30 min at 37 °C in the dark. Cells were then washed, and DCF fluorescence was determined using an LSRFortessa (BD Biosciences). The assessment was repeated 3 times in independent experiments. Quantitative data and figures were obtained using FlowJo 7.6.2 Data Analysis Software. The assessment was repeated three times in independent experiments blind to experimenter and flow cytometer analyst.

**Analysis of mitochondrial membrane potential (ΔΨm) by fluorescence microscopy.** The hWJ-MSCs or ChLNs were left in regular culture medium (RCm) for 0, 2 and 4 days. Then, the cells (5 x $10^3$) were incubated with the passively diffusing and active mitochondria-accumulating dye deep-red MitoTracker compound (20 nM, final concentration) for 20 min at RT in the dark (Invitrogen, cat # M22426). Cells were then washed twice with PBS. Mito-Tracker fluorescence intensity was determined by analysis of fluorescence microscopy images. The assessment was repeated three times in independent experiments. The nuclei were stained with 0.5 μM Hoechst 33342 (2.5 μM) staining compound. The assessment was repeated three times in independent experiments blind to experimenter and flow cytometer analyst.

**Analysis of mitochondrial membrane potential (ΔΨm) by flow cytometry.** ChLNs were left in regular medium for 0, 2 and 4 days. Then, the cells ($1\times10^5$) were incubated for 30 min at RT in the dark with MitoTracker (20 nM, final concentration). The cells were analyzed using an LSRFortessa (BD Biosciences). The experiment was performed three times in independent experiments, and 10,000 events were acquired for analysis. Quantitative data and figures were obtained using FlowJo 7.6.2 Data Analysis Software. The assessment was repeated three times in independent experiments blind to experimenter and flow cytometer analyst.

**Determination of DNA fragmentation by flow cytometry.** DNA fragmentation was determined using a hypotonic solution of PI. Cells entering the cellular cycle phase sub-$G_0$ were identified as those undergoing apoptosis. ChLNs were left in regular medium for 0, 2 and 4 days. Then, the cells ($1x10^5$) were detached, washed twice with PBS (pH 7.2) and stored in 95% ethanol overnight at -20 °C. The cells were washed and incubated in 400 μL solution containing propidium iodide (PI; 50 μg/ml), RNase A (100 μg/ mL), EDTA (50 mM), and Triton X-100 (0.2%) for 60 min at 37 °C. The cell suspension was analyzed for PI fluorescence using an Epics XL flow cytometer (Beckman Coulter). DNA fragmentation was assessed 3 times in independent experiments. Quantitative data and figures from the sub-$G_0$/$G_1$ population were obtained using FlowJo 7.6.2 Data Analysis Software. The assessment was repeated three times in independent experiments blind to experimenter and flow cytometer analyst.

**Acetylcholinesterase activity measurement.** We determined the acetylcholinesterase (AChE) activity using the AChE Assay Kit (Abcam, Cat# ab138871) according to the manufacturer's protocol. Briefly, ChLNs at days 0, 2 and 4 of post differentiation were detached with 0.25% trypsin and mechanically lysed by freezing/sonication. Lysates were centrifuged at 18,894 x *g* for 15 min, and supernatants were used for protein quantification by the BCA method (see above) and the detection of AChE activity. AChE degrades the neurotransmitter acetylcholine (ACh) into choline and acetic acid. We used the DTNB (5,5′-dithiobis(2-nitro-benzoic acid)) reagent to quantify the thiocholine produced from the hydrolysis of acetylthiocholine by AChE. The absorption intensity of the DTNB adduct was used to measure the amount of thiocholine formed, which was proportional to AChE activity. We read the absorbance in a microplate reader at ~410 nm. The data obtained were compared to the standard curve values, and the AChE amounts (mU) were normalized to protein values (mU/ mg protein). The assessment was repeated three times in independent experiments blind to experimenter.

**Measurement of Aβ$_{1-42}$ peptide in culture medium.** The level of Aβ$_{1-42}$ peptide was measured according to a previous report [39] with minor modifications. Briefly, WT and PSEN1 E280A ChLNs were left in regular medium for 0, 2 and 4 days. Then, 100 μl of conditioned medium was collected, and the levels of secreted Aβ$_{1-42}$ peptides were determined by a solid-phase sandwich ELISA (Invitrogen, Cat# KHB3544) following the manufacturer's instructions. The assessment was repeated three times in independent experiments blind to experimenter.

**Intracellular calcium imaging.** Intracellular calcium (Ca$^{2+}$) concentration changes evoked by cholinergic stimulation were assessed according to refs. [40, 41] with minor modifications. For the measurement, the fluorescent dye Fluo-3 (Fluo-3 AM; Thermo Fisher Scientific, cat: F1242) was employed. The dye was dissolved in DMSO (1 mM) to form a stock solution. Before the experiments, the stock solution was diluted in neuronal buffer solution (NBS buffer: 137 mM NaCl, 5 mM KCl, 2.5 mM CaCl$_2$, 1 mM MgCl$_2$, pH 7.3, and 22 mM glucose). The working concentration of the dye was 2 μM. The WT and PSEN1 E280A ChLNs were incubated for 30 min at 37 ˚C with the dye-containing NBS and then washed five times. Intracellular Ca$^{2+}$ transients were evoked by acetylcholine (1 mM final concentration) at 0, 2 and 4 days post differentiation. The measurements were carried out using the 100x objective of the microscope. Several regions of interest (ROIs) were defined in the visual field of the camera. One of the ROIs was cell-free, and the fluorescence intensity measured here was considered background fluorescence (F$_{bg}$). The time dependence of the fluorescence emission was acquired, and the fluorescence intensities (hence the Ca$^{2+}$ levels) were represented by pseudo-colors. To calculate the changes of the average Ca$^{2+}$-related fluorescence intensities, the F$_{bg}$ value was determined from the cell-free ROI, and then the resting fluorescence intensities (F$_{rest}$) of the cell-containing ROIs were obtained as the average of the points recorded during a period of 10 s prior to the addition of acetylcholine. The peaks of the fluorescence transients were found by calculating the average of four consecutive points and identifying those points that gave the highest average value (F$_{max}$). The amplitudes of the Ca$^{2+}$-related fluorescence transients were expressed relative to the resting fluorescence (ΔF/F) and were calculated by the following formula: ΔF/F = (F$_{max}$-F$_{rest}$)/(F$_{rest}$-F$_{bg}$). For the calculation of the fluorescence intensities, ImageJ was used. The terms fluorescence intensity was used as an indirect indicator of intracellular Ca$^{2+}$ concentration. The assessment was repeated three times in independent experiments blind to experimenter.

**JNK inhibition experiment.** The ChLNs were left in regular medium for 0, 2 and 4 days alone or co-incubated with the anthrapyrazolone JNK inhibitor SP600125 (1 μM final concentration). This compound competes with ATP to inhibit the phosphorylation of c-JUN. After this time, cells were evaluated for p-TAU and t-TAU protein expression by immunofluorescence, as described above. The assessment was repeated three times in independent experiments blind to experimenter.

## Photomicrography and image analysis

Light microscopy photographs were taken using a Zeiss Axiostart 50 Fluorescence Microscope equipped with a Canon PowerShot G5 digital camera (Zeiss Wöhlk-Contact-Linsen, Gmb Schcönkirchen, Germany), and fluorescence microscopy photographs were taken using a Zeiss Axiostart 50 Fluorescence Microscope equipped with a Zeiss AxioCam Cm1 and (Zeiss Wöhlk-Contact-Linsfluoreen, Gmb Schcönkirchen, Germany) and Floyd Cells Imaging Station microscope. Fluorescence images were analyzed by ImageJ software (http://imagej.nih.gov/ij/). The figures were transformed into 8-bit images, and the background was subtracted. The cellular measurement regions of interest (ROIs) were drawn around the nucleus (for the

case of transcription factors and apoptosis effectors) or over all cells (for cytoplasmic probes), and the fluorescence intensity was subsequently determined by applying the same threshold for cells in the control and treatment conditions. Mean fluorescence intensity (MFI) was obtained by normalizing total fluorescence to the number of nuclei.

## Data analysis

In this experimental design, a vial of MSCs was thawed, cultured and the cell suspension was pipetted at a standardized cellular density of 2.6 x $10^4$ cells/cm$^2$ into different wells of a 24-well plate. Cells (i.e., the biological and observational unit [42]) were randomized to wells by simple randomization (sampling without replacement method), and then wells (i.e., the experimental units) were randomized to treatments by similar method. Experiments were conducted in triplicate wells. The data from individual replicate wells were averaged to yield a value of n = 1 for that experiment and this was repeated on three occasions blind to experimenter and/ or flow cytometer analyst for a final value of n = 3 [42]. Based on the assumptions that the experimental unit (i.e. the well) data comply with independence of observations, the dependent variable is normally distributed in each treatment group (Shapiro-Wilk test), and there is homogeneity of variances (Levene's test), the statistical significance was determined by a One-way analysis of variance (ANOVA) followed by Tukey's post hoc comparison calculated with GraphPad Prism 5.0 software. Differences between groups were only deemed significant when a p-value of < 0.05 (*), < 0.001 (**) and < 0.001 (***). All data are illustrated as the mean ± S.D.

## Results

### PSEN1 E280A and wild-type WJ-MSCs show similar phenotype, immunophenotype, lineage differentiation and transdifferentiation

We first evaluated whether the PSEN1 E280A mutation affects the capacity of MSCs to generate mesodermal and ectodermal lineages through differentiation and transdifferentiation, respectively. Therefore, WJ-MSCs were isolated from ten umbilical cords of volunteers according to standard procedures [43], and PCR-RFLP electrophoretic profile analysis identified one umbilical cord sample out of ten as a carrier of the mutation *PSEN1* c.839A > C, p.E280A (PSEN1 E280A MSCs, S1 Fig). Then, PSEN1 E280A MSCs and wild-type WJ-MSCs (WT PSEN1 MSCs) were further cultured and characterized for morphological, karyotype, immuno-phenotypic features and differentiation capabilities. As shown in Fig 1, WT PSEN1 and PSEN1 E280A MSCs displayed the typical colony-forming units (Fig 1A and 1B), adherent growth, and fibroblast-like cellular morphology (Fig 1C and 1D). Karyotype analysis showed no chromosomal alterations (Fig 1E and 1F), and APOE genotyping analysis showed that PSEN1 E280A cells had the APOE*3/4 genotype and WT PSEN1 cells had the APOE*3/3 genotype (S2 Fig). Flow cytometry analysis showed that both wild-type and mutant MSCs were positive (>95% of positive cells) for mesenchymal associated markers CD73, CD90 and CD9 (Fig 1G and 1H) but negative (<5% of positive cells) for hematopoietic cell surface antigens CD34/ CD45. WT and PSEN1 E280A MSCs cultured in adipogenic, osteogenic, or chondrogenic induction medium differentiated into adipocytes (Fig 1J and 1P), osteoblasts (Fig 1L and 1R), and chondrocytes (Fig 1N and 1T), respectively, while MSCs cultured in regular culture medium were undifferentiated (Fig 1I, 1O, 1K, 1Q, 1M and 1S).

Additionally, MSCs were transdifferentiated into cholinergic-like neurons (ChLNs) from WJ-MSCs using a new method [29]. As shown in Fig 2, WT PSEN1 and PSEN1 E280A WJ-MSCs cultured in minimal culture medium (MCm) for 7 days expressed basal levels of protein MAP2 (Fig 2A and 2B) and β Tub III (Fig 2A and 2C) and undetectable levels of

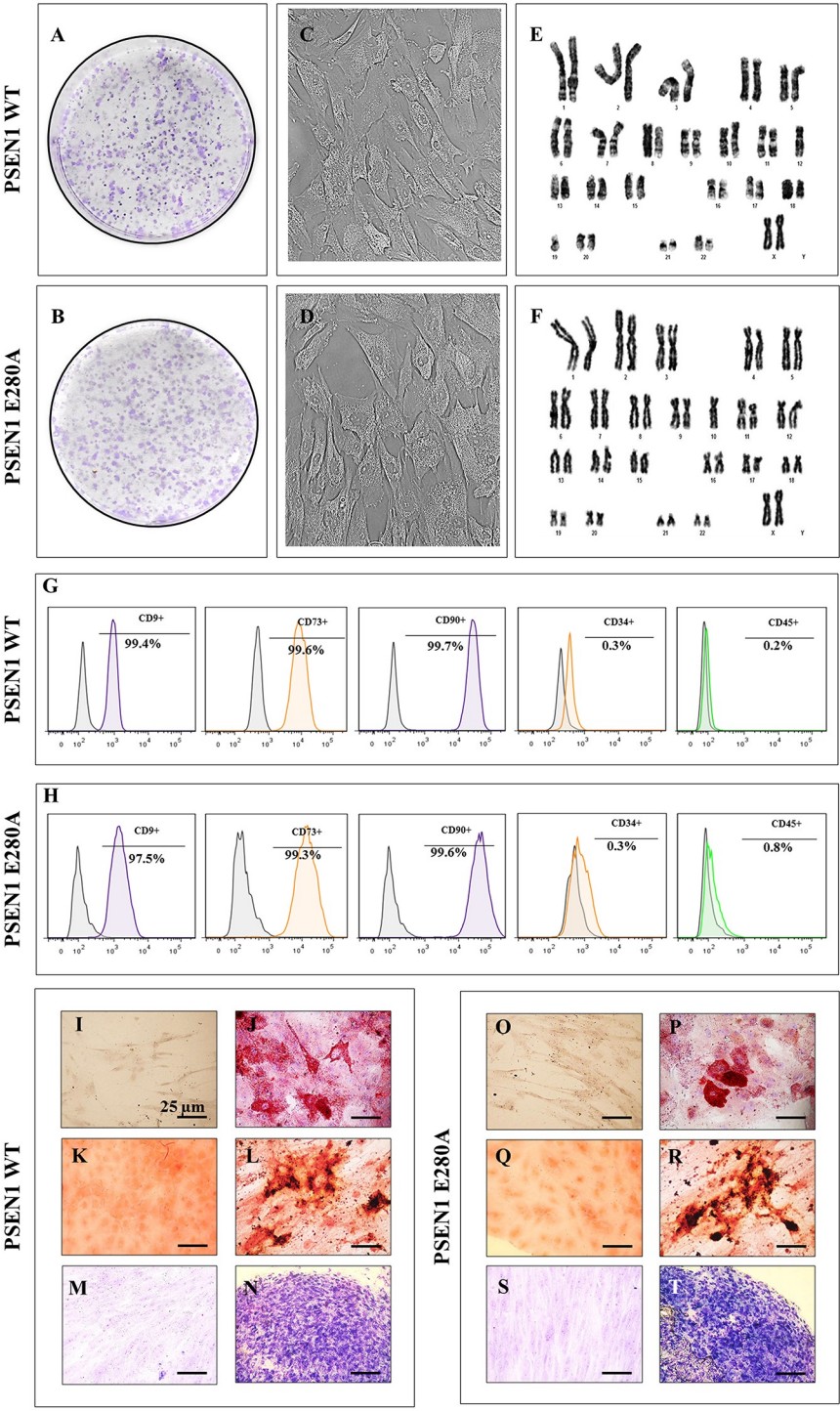

**Fig 1. Characterization of WT PSEN1 and PSEN1 E280A WJ-MSCs. (A-B)** Representative images showing the colony-forming units and **(C-D)** adherent growth and fibroblast-like morphology typical of WJ-MSCs. **(E-F)** Karyotype analysis performed at passage 2 showing chromosomal normality (46XX). **(G-H)** Flow cytometry analyses showing the percentage of double-positive CD9/ CD73/ CD90/ CD34 and CD45 WJ-MSCs. **(I, O)** Oil-Red-O negatively stained undifferentiated WJ-MSCs grown on regular culture medium. **(J, P)** Oil-Red-O positively stained adipocytes differentiated from WJ-MSCs showing intracellular red lipidic vacuoles. **(K, Q)** Von Kossa negatively stained undifferentiated WJ-MSCs grown on regular culture medium. **(L, R)** Von Kossa positively stained osteoblasts differentiated from WJ-MSCs showing silver intracellular precipitates. **(M, S)** Toluidine blue negatively stained undifferentiated WJ-MSCs grown on regular culture medium. **(N, T)** Toluidine blue positively stained chondrocytes differentiated from WJ-MSCs showing extracellular glycoprotein matrix. Image magnification, 400x. The images represent 1 out of 3 independent experiments.

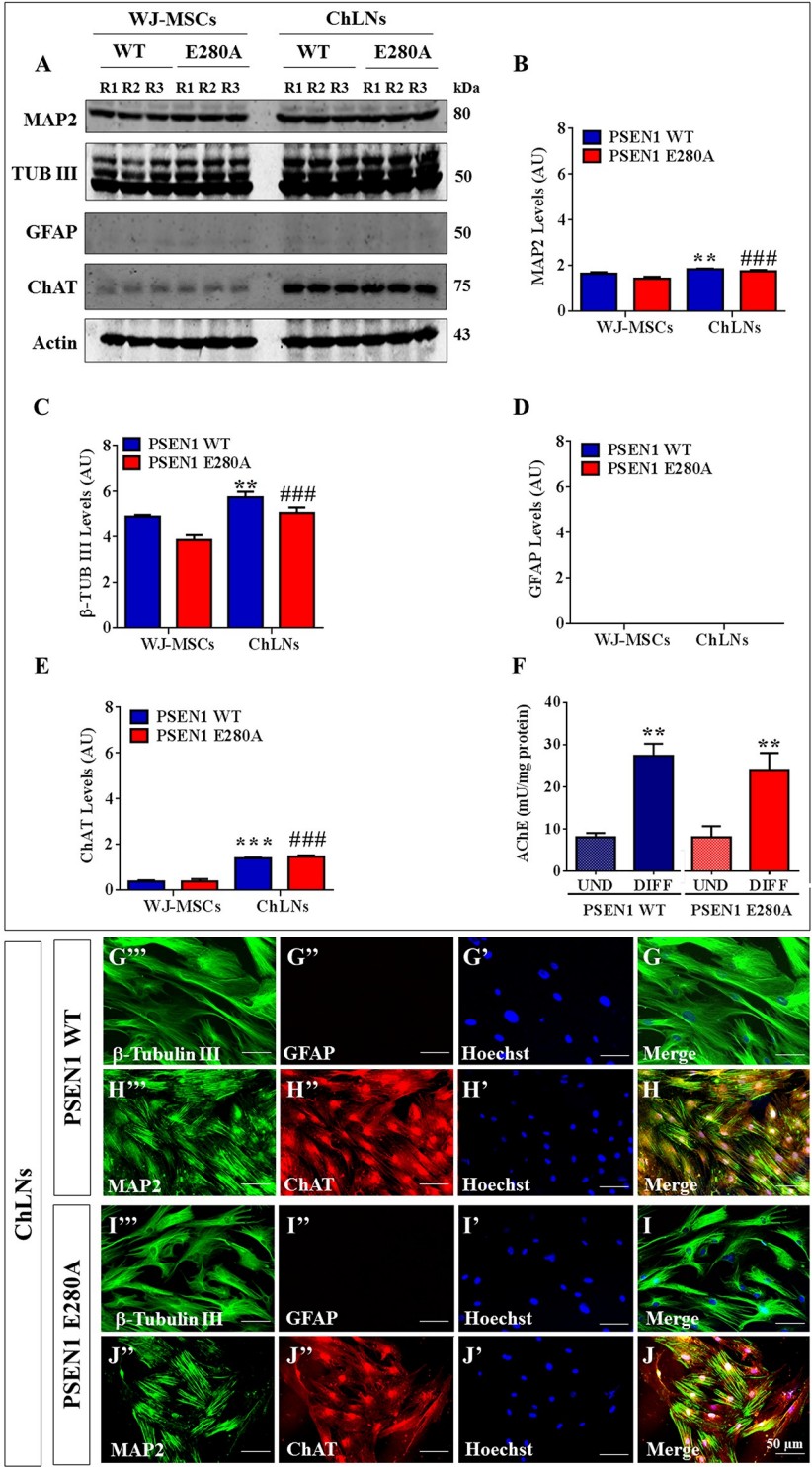

**Fig 2. Cholinergic-like neurons differentiate from WT PSEN1 and PSEN1 E280A WJ-MSCs.** WT PSEN1 and PSEN1 E280A WJ-MSCs were cultured in cholinergic differentiation medium as described in the *Materials and Methods* section for 7 days. After this time, the proteins in the extracts were blotted with primary antibodies against MAP2, β-tubulin III, GFAP, ChAT and actin proteins. The intensities of the western blot bands shown in (**A**) were measured (**B, C, D, E**) by an infrared imaging system (Odyssey, LI-COR), and the intensity was normalized to that of actin. (**F**) Measurements of acetylcholinesterase activity in WT PSEN1 and PSEN1 E280A ChLNs after 7 days of transdifferentiation. (**G-J**) Cells were double stained as indicated in the figure with primary antibodies against β-

tubulin III (**green; G'" and I"**) and GFAP (**red; G" and I"**) or MAP2 (**green; H'" and J'"**) and ChAT (**red; H" and J"**). The nuclei were stained with Hoechst 33342 (**blue; G'-J'**). Data are expressed as the mean ± SD; *$p<0.05$; **$p<0.01$; ***$p<0.001$. The blots and figures represent 1 out of 3 independent experiments. Image magnification, 200x.

GFAP (Fig 2A and 2D) and ChAT (Fig 2A and 2E). As expected, when the cells were exposed to cholinergic-N-Run medium (Ch-N-Rm) for 7 days [29], the levels of protein MAP2 (Fig 2A and 2B), β Tub III (Fig 2A and 2C) and ChAT (Fig 2A and 2E) were significantly higher than those in cells exposed to MCm. Noticeably, ChLNs remained negative for the specific glial cell lineage marker GFAP (Fig 2A and 2D). These observations were confirmed by immunofluorescence (Fig 2G–2J). Because the enzyme AChE catalyzes the breakdown of the neurotransmitter acetylcholine (ACh, [44]), we evaluated whether ChLNs expressed a catalytically functional AChE enzyme. As shown in Fig 2F, the AChE enzyme presented basal activity in WJ-MSCs under MCm culture conditions (~8 ± 1.2 mU/mg protein), while the AChE enzyme activity was significantly higher (at least a 3-fold increase) in both control and mutant ChLNs (~26 ± 1.6 mU/mg protein). Remarkably, there was no significant difference in AChE enzymatic activity ($p< 0.05$) between WT and PSEN1 E280A ChLNs (at day 0 of post differentiation).

## WT and PSEN1 E280A WJ-MSCs show similar levels of intracellular APP / Aβ$_{42}$, oxidized DJ-1, mitochondrial membrane potential (ΔΨ$_m$) and Reactive Oxygen Species (ROS)

Next, we evaluated whether the PSEN1 E280A mutation induced overproduction of APP/ Aβ$_{42}$ and OS, and alterations in ΔΨ$_m$ and ROS production in MSCs. After 7 days of culture in MCm, MSCs were left in RCm for 0, 2 and 4 additional days. Western blot measurements and immunofluorescence analysis revealed that both WT and PSEN1 E280A MSCs displayed undetectable levels of intracellular APP / Aβ$_{42}$ and oxidized DJ-1 at days 0, 2 and 4 (S3A–S3J Fig). Moreover, both WT and PSEN1 E280A MSCs displayed normal ΔΨ$_m$ (S4A Fig), undetectable levels of ROS (S4B Fig) and no changes in DNA content (S4C Fig) at days 0, 2 and 4 in RCm according to flow cytometry (S4A–S4C Fig) and DCF/ MitoTracker fluorescence (S4D–S4K Fig).

## PSEN1 E280A Cholinergic-Like Neurons (ChLNs) show high levels of intracellular sAPPβf, oxidized DJ-1, reactive oxygen species (ROS), loss of mitochondrial membrane potential (ΔΨ$_m$) and DNA fragmentation, but none of those markers are detected in WT PSEN1 ChLNs

The above observations prompted us to evaluate the same cell parameters in ChLNs. We initially verify that the antibody clone 6E10 recognizes Aβ monomers as well as large aggregates of Aβ$_{42}$ and APP. Effectively, the antibody recognized the "unaggregated" (monomers), and large aggregates forms of synthetic Aβ$_{42}$ peptide (Fig 3A). The WT PSEN1 and PSEN1 E280A ChLNs (obtained after 7 days in Ch-N-Rm) were left in regular culture medium (RCm) for 0, 2 and 4 additional days post transdifferentiation (hereafter labelled as 0, 2, or 4). Western blot also revealed that WT PSEN1 ChLNs displayed undetectable levels of intracellular APP/ Aβ$_{42}$ aggregates (Fig 3A and 3B) and oxidized DJ-1 (Fig 3A and 3C), whereas flow cytometry showed no loss of ΔΨ$_m$ (Fig 4A–4C) and no ROS generation (Fig 4D–4F) at any time tested. However, PSEN1 E280A ChLNs exhibited significantly higher levels of intracellular APP/ Aβ$_{42}$ aggregates (Fig 3A and 3B) and oxidized DJ-1 (Fig 3A and 3C) and lower ΔΨ$_m$ than WT PSEN1 ChLNs at days 2 and 4 (Fig 4B and 4C). These observations were confirmed by

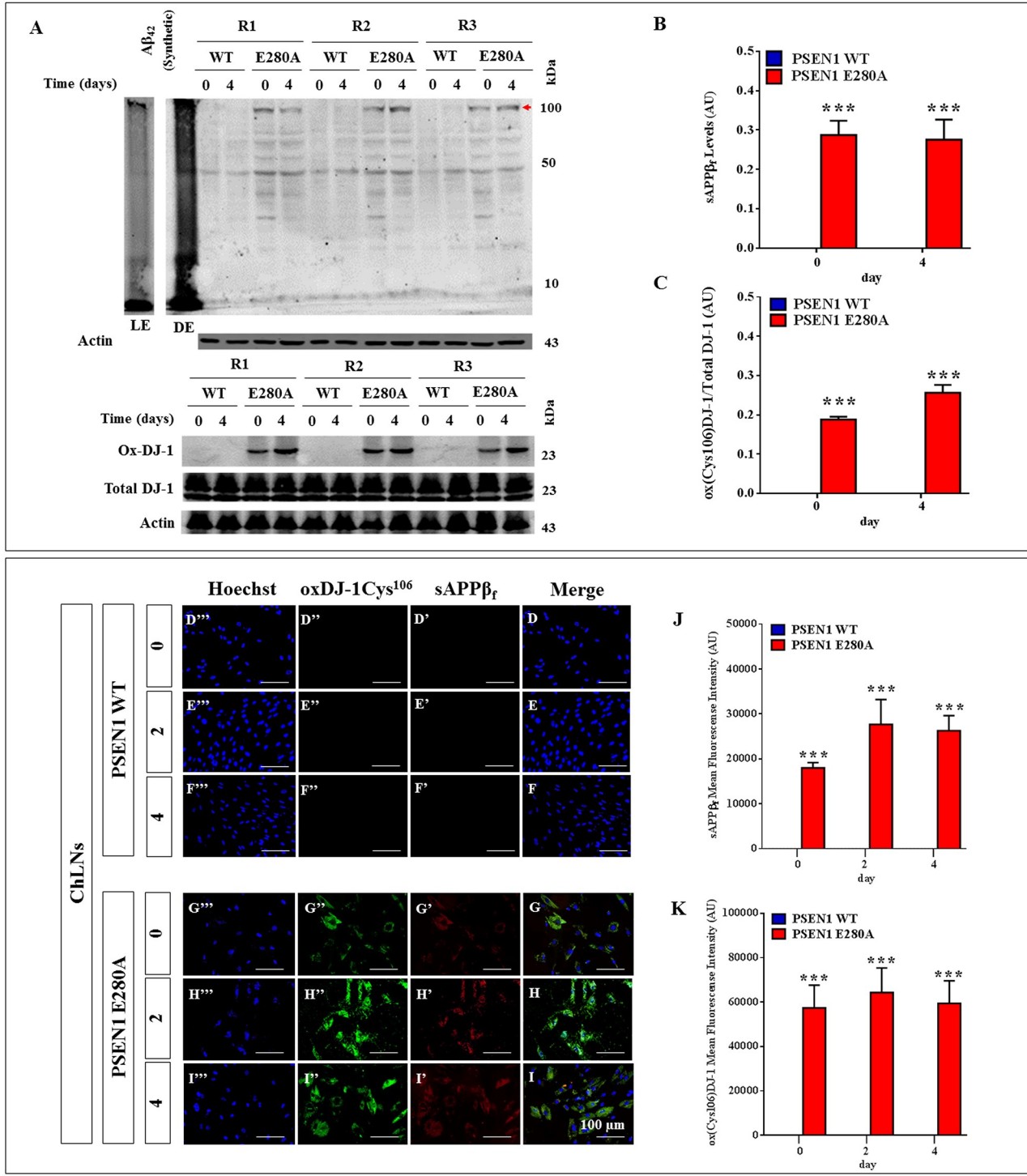

**Fig 3. PSEN1 E280A Cholinergic-Like Neurons (ChLNs) show high levels of intracellular sAPPβf and oxidized DJ-1.** Synthetic unaggregated (i.e., monomers) and large oligomeric Aβ42 assemblies were prepared as described in *Material and Methods* section according to Ref. [38]. The ChLNs were obtained as follows. After 7 days of transdifferentiation, WT PSEN1 and PSEN1 E280A ChLNs were left in regular culture medium (RCm) for 0 and 4 days, as indicated in the figure. After this time, the proteins in the extracts were blotted with primary antibodies against Aβ42, oxDJ-1Cys106 and actin proteins. The intensities of the western blot bands shown in **(A)** were measured **(B, C)** by an infrared imaging system (Odyssey, LI-COR), and the intensity was normalized to that of actin. Additionally, cells were double stained as indicated in the figure **(D-I)** with primary antibodies against APP751/ Aβ42 (*red*; **D'-I'**) and oxDJ-1Cys106 (*green*; **D"- I"**). The nuclei were stained with Hoechst 33342 (*blue*; **D'"- I'"**). **(J)** Quantification of Aβ42 fluorescence intensity. **(K)** Quantification of oxDJ-1Cys106 fluorescence intensity. Data are expressed as the mean ± SD; $^{*}p<0.05$; $^{**}p<0.01$; $^{***}p<0.001$. The blots and figures represent 3 independent experiments (Repeat 1, R2, and R3). The *arrowhead* represents APP and/or Aβ42 aggregates. Image magnification, 200x; inset magnification, 800x. LE = light exposure; DE = dark exposure.

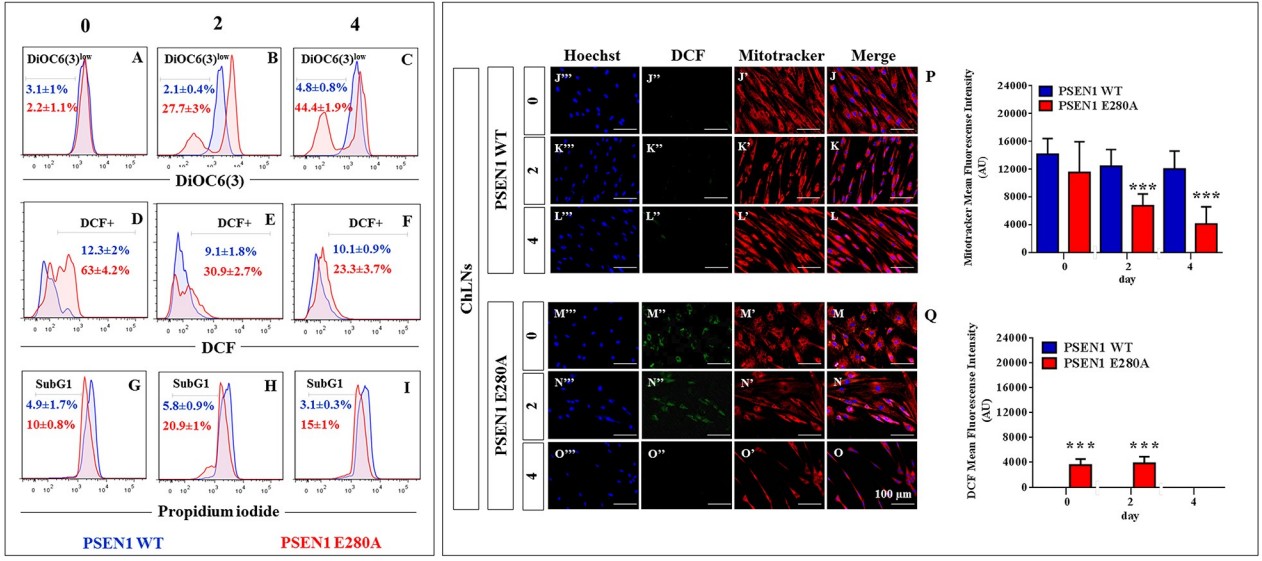

**Fig 4. PSEN1 E280A Cholinergic-Like Neurons (ChLNs) show loss of mitochondrial membrane potential ($\Delta\Psi_m$), high levels of intracellular reactive oxygen species (ROS), and fragmentation of DNA.** After 7 days of transdifferentiation, WT PSEN1 and PSEN1 E280A ChLNs were left in regular culture medium (RCm) for 0, 2 and 4 days, as indicated in the figure. Representative histograms showing $DiOC_6(3)^{low}$ **(A-C)**, DCF+ **(D-F)** and $SubG_1$ **(G-I)** populations from WT PSEN1 (*blue*) and PSEN1 E280A (*Red*) ChLNs. Representative MitoTracker **(J'-O')**, DCF **(J''-O'')**, Hoechst **(J'''-O''')** and merge **(J-O)** pictures of WT PSEN1 and PSEN1 E280A ChLNs. **(P)** Quantification of MitoTracker fluorescence intensity. **(Q)** Quantification of DCF fluorescence intensity. Data are expressed as the mean ± SD; $^*p<0.05$; $^{**}p<0.01$; $^{***}p<0.001$. The histograms and figures represent 1 out of 3 independent experiments. Image magnification, 200x.

immunofluorescence microscopy (Figs 3D–3K and 4J'–4O'). To confirm the identity of the intracellular APP/ $A\beta_{42}$ aggregates displayed in the Western blot (Fig 3A, *arrowhead*), we used LC-MS/MS technique. The LC-MS/MS analysis identified two peptides $^{364}$LPTTAASTP-DAVDK$^{377}$ (S5A Fig) and $^{430}$AVIQHFQEK$^{438}$ (S5B Fig) that showed high homology with APP fragments (e.g., APP714, APP733, APP751, APP752, S5C Fig), hereafter sAPPβf. No $A\beta_{42}$ fragment was identified in the interrogated gel region. Interestingly, PSEN1 E280A ChLNs showed high levels of ROS production as early as day 0, that is, at day 7 of transdifferentiation (Fig 4D, 4M'' and 4Q) and at day 2 post transdifferentiation (Fig 4E, 4N'' and 4Q), but ROS were severely reduced at day 4 (Fig 4F, 4O'' and 4Q). Furthermore, both WT and PSEN1 E280A ChLNs exhibited a typical quiescent cell cycle (i.e., phase $G_0$, Fig 4G–4I) and showed similar numbers of cells in the $SubG_0$ population at day 0 (Fig 4G); nonetheless, the $SubG_0$ population was significantly elevated in PSEN1 E280A ChLNs at day 2 (~15% compared to that in WT cells; Fig 4H) and day 4 (~12% compared to WT cells; Fig 4I) according to propidium iodide (PI) fluorescence analysis.

## PSEN1 E280A ChLNs display activation of p53, PUMA, c-JUN and CASPASE-3

Next, we wanted to investigate whether the intracellular sAPPβf induced cell death signaling in PSEN1 ChLNs. Therefore, we used the transcription factors p53 and c-JUN, pro-apoptotic BH3-only protein PUMA, and protease CASPASE-3 as cell death markers to examine the pro-death activity of sAPPβf in ChLNs over time. As shown in Fig 5, while WT PSEN1 ChLNs showed no detectable levels of apoptogenic proteins at any time tested (Fig 5A–5E), the PSEN1 E280A ChLNs displayed significant detection of c-JUN (at day 0, 2 and 4, Fig 5A and 5B), p53 (at day 2 and 4, Fig 5A and 5C), PUMA (at day 4, Fig 5A and 5D) and CASP-3 (at day 4,

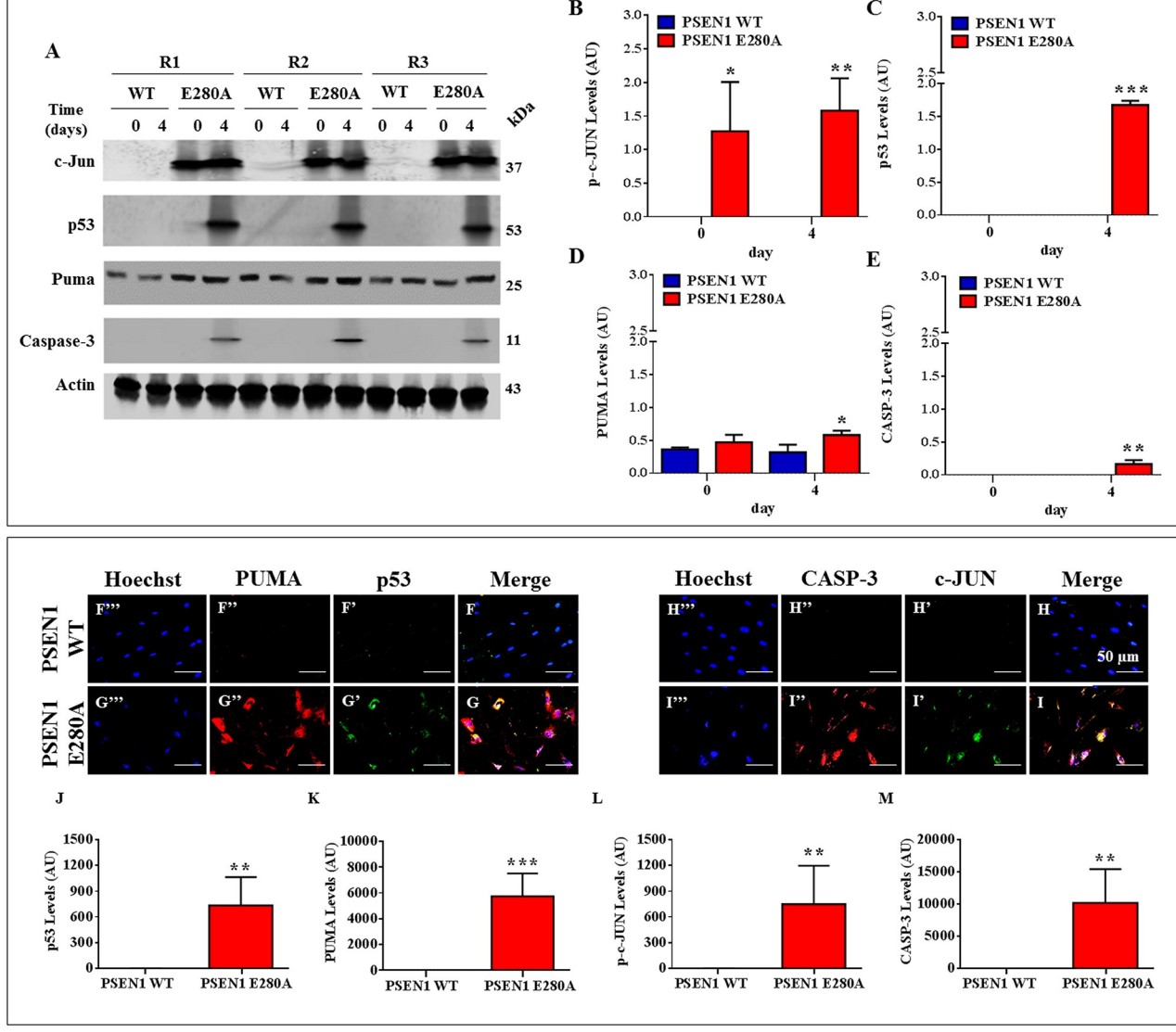

**Fig 5. PSEN1 E280A ChLNs display activation of p53, PUMA, c-JUN and CASPASE-3.** After **7** days of transdifferentiation, WT PSEN1 and PSEN1 E280A ChLNs were left in regular culture medium (RCm) for 0, 2 and 4 additional days, as indicated in the figure. After this time, the proteins in the extracts were blotted with primary antibodies against phosphorylated c-JUN (c-JUN), p53, PUMA, CASPASE-3 (CASP-3) and actin proteins. The intensities of the western blot bands shown in (**A**) were measured (**B, C, D and E**) by an infrared imaging system (Odyssey, LI-COR), and the intensity was normalized to that of actin. Additionally, after 4 days, ChLNs were double stained as indicated in the figure (**F-I**) with primary antibodies against p53 (*green*; **F'** and **G'**), PUMA (*red*; **F"** and **G"**), c-JUN (*green*; **H'** and **I'**) and CASP-3 (*red*; **H"** and **I"**). The nuclei were stained with Hoechst 33342 (*blue*; **F'''-I'''**). (**J-M**) Quantification of p53 (**J**), PUMA (**K**), c-JUN (**L**) and CASP-3 (**M**) fluorescence intensity. Data are expressed as the mean ± SD; $^{*}p<0.05$; $^{**}p<0.01$; $^{***}p<0.001$. The blots and figures represent 1 out of 3 independent experiments. Image magnification, 200x.

Fig 5A and 5E). These observations were confirmed by immunofluorescence analysis at day 4 (Fig 5F–5M).

## PSEN1 E280A ChLNs show reduced functional response to Acetylcholine (ACh), low acetylcholinesterase activity, and generation of high levels of extracellular Aβ₄₂ peptide

We further investigated whether WT and PSEN1 E280A ChLNs responded to ACh stimuli as an evaluation of cholinergic neuronal $Ca^{2+}$ responsiveness and functionality [29]. To this aim,

we simultaneously measured the secreted levels of $A\beta_{42}$ and evaluated the response of ChLNs to ACh. ELISA revealed that both WT PSEN1 and PSEN1 E280A ChLNs secreted similarly low levels of extracellular $A\beta_{42}$ at days 0 and 2, but at day 4, PSEN1 E280A ChLNs showed significantly higher extracellular levels of $A\beta_{42}$ compared to PSEN1 WT ChLNs (~3.2-f.i., Fig 6A). For functional analysis, both control and mutant PSEN1 ChLN cultures were puffed with ACh (1 mM final concentration) into a bath solution. As expected, ACh induced a transient elevation of intracellular $Ca^{2+}$ in WT PSEN1 ChLNs at day 0 (Fig 6B and 6D), 2 (Fig 6E and 6G) and 4 post-transdifferentiation (Fig 6H and 6J). The average fluorescence change ($\Delta F/F$) was $6.4 \pm 0.6$, $6.4 \pm 0.7$ and $5.9 \pm 1.4$–fold, respectively, with a mean duration of $40 \pm 10$ s ($n = 20$ ChLN cells imaged, $N = 3$ dishes) according to cytoplasmic $Ca^{2+}$ responses to Fluo-3-mediated imaging. Interestingly, PSEN1 E280A ChLNs showed higher intracellular $Ca^{2+}$ in response to ACh treatment at day 0 (Fig 6C and 6D) and 2 (Fig 6F and 6G) than WT PSEN1 ChLNs (Fig 6D and 6G). The average fluorescence change ($\Delta F/ F$) was $4.6 \pm 1.2$ and $5.1 \pm 1.1$–fold, respectively, with a mean duration of $40 \pm 10$ s ($n = 20$ ChLN cells imaged, $N = 3$ dishes). Nevertheless, we found a significant difference in the $\Delta F/ F$ response ($\Delta F/ F = 1.1 \pm 0.75$, $p < 0.001$) in PSEN1 E280A ChLNs exposed to ACh at 4 days post differentiation (Fig 6I and 6J) compared to that in WT ChLNs (Fig 6H and 6J).

The above findings compelled us to evaluate whether ChLNs expressed a catalytically functional acetylcholinesterase (AChE) enzyme at similar times post of transdifferentiation. As shown in Fig 6K, the AChE enzyme showed similar catalytic activity at days 2 and 4 in both WT PSEN1 and PSEN1 E280A ChLNs. However, the AChE enzyme activity in PSEN1 E280A ChLNs was significantly lower (at least 5-fold) than that in WT PSEN1 ChLNs at day 4.

## PSEN1 E280A ChLNs induce phosphorylation of TAU protein

Several experimental data support $A\beta$-induced TAU pathology (e.g., ref. [45]), however no data is available to determine whether sAPPβf can induce protein TAU phosphorylation. We thus evaluated whether PSEN1 E280A ChLNs display abnormal levels of phosphorylated TAU protein at residues $Ser^{202}$ and $Thr^{205}$, two well-known hyperphosphorylated epitopes involved in AD pathology [46]. To this end, wild-type and mutant ChLNs were left in regular culture medium (RCm) for 0, 2 and 4 days. Then, the ratio of phosphorylated TAU (p-TAU)/total TAU (t-TAU) was determined by western blot and immunofluorescence imaging analyses. As shown in Fig 7, while WT PSEN1 ChLNs showed no phosphorylation of TAU protein according to western blot (Fig 7A and 7B) and immunofluorescence microscopy (Fig 7C'–7E' and 7I), PSEN1 E280A ChLNs presented an increase in the p-TAU/ t-TAU ratio at 4 days post transdifferentiation (Fig 7A, 7B, 7F'–7H' and 7I). JNK has been suggested to phosphorylate TAU at the Ser202/Thr205 epitopes [47]. Therefore, we determined whether JNK was implicated in TAU phosphorylation in ChLNs. To this end, PSEN1 E280A ChLNs were incubated with the JNK inhibitor SP600125 for 0, 2 and 4 days. Notably, the p-TAU/ t-TAU ratio remained unaltered at 0 (Fig 7J, 7K, 7O' and 7R), 2 (Fig 7J, 7K, 7P' and 7R) and 4 days post differentiation (Fig 7J, 7K, 7Q' and 7R).

## Discussion

Currently, the neuropathology of AD includes extracellular deposits of $A\beta$ in plaques, intracellular neurofibrillary tangles comprising hyperphosphorylated TAU, synaptic dysfunction, and neuronal death. In an effort to explain such observations, several theories have been proposed [48]; however, the amyloid cascade hypothesis has prevailed for more than 25 years [49]. The $A\beta$ hypothesis postulates that an imbalance in the production of extracellular Aβ42 plaques by mutations in at least three genes (e.g., *APP*, *PSEN1; PSEN2*) is an early initiating factor in AD.

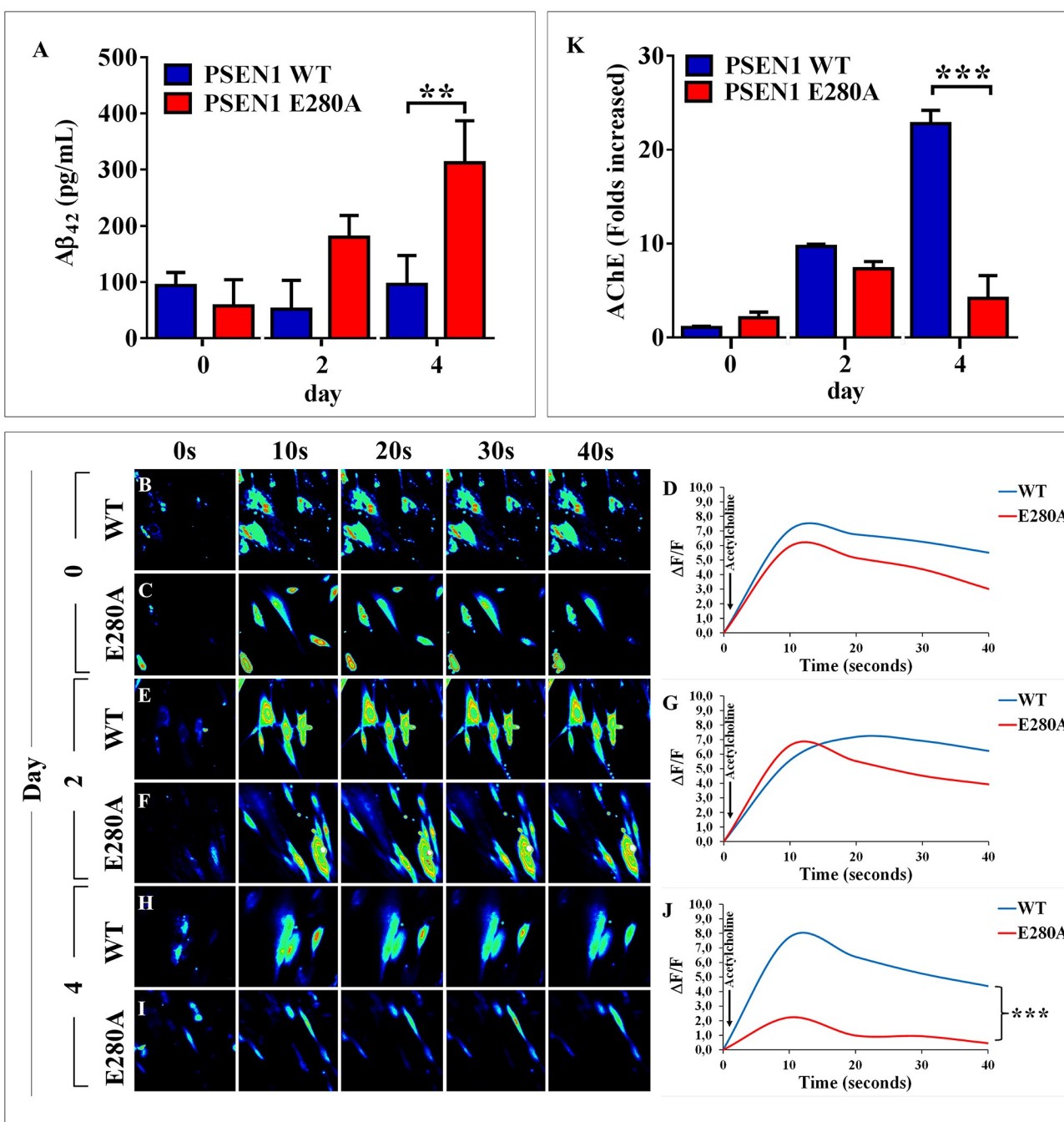

**Fig 6. PSEN1 E280A ChLNs show a reduced functional response to Acetylcholine (ACh), low acetylcholinesterase activity and high levels of extracellular Aβ₄₂ peptide.** After 7 days of transdifferentiation, WT PSEN1 and PSEN1 E280A ChLNs were left in regular culture medium for 0, 2 and 4 days, as indicated in the figure. **(A)** ELISA quantification of extracellular Aβ₄₂ peptide in supernatants. Time-lapse images (0, 10, 20, 30, 40 s) of $Ca^{2+}$ fluorescence in WT PSEN1 and PSEN1 E280A ChLNs after 0 **(B-C)**, 2 **(E-F)** and 4 days **(H-I)** in response to ACh treatment. ACh was puffed into the culture at 10 s (*arrow*). Then, the $Ca^{2+}$ fluorescence of the cells was monitored at the indicated times. Color contrast indicates fluorescence intensity: dark blue < light blue < green < yellow < red. **(D, G, J)** Normalized mean fluorescence signal (ΔF/F) over time from the cells indicating temporal cytoplasmic $Ca^{2+}$ elevation in response to ACh treatment. **(K)** Measurement of acetylcholinesterase activity at 0, 2 and 4 days post transdifferentiation. Data are presented as the means ± SD. $^{*}p<0.05$; $^{**}p<0.01$; $^{***}p<0.001$. The histograms and figures represent 1 out of 3 independent experiments. Image magnification, 400x.

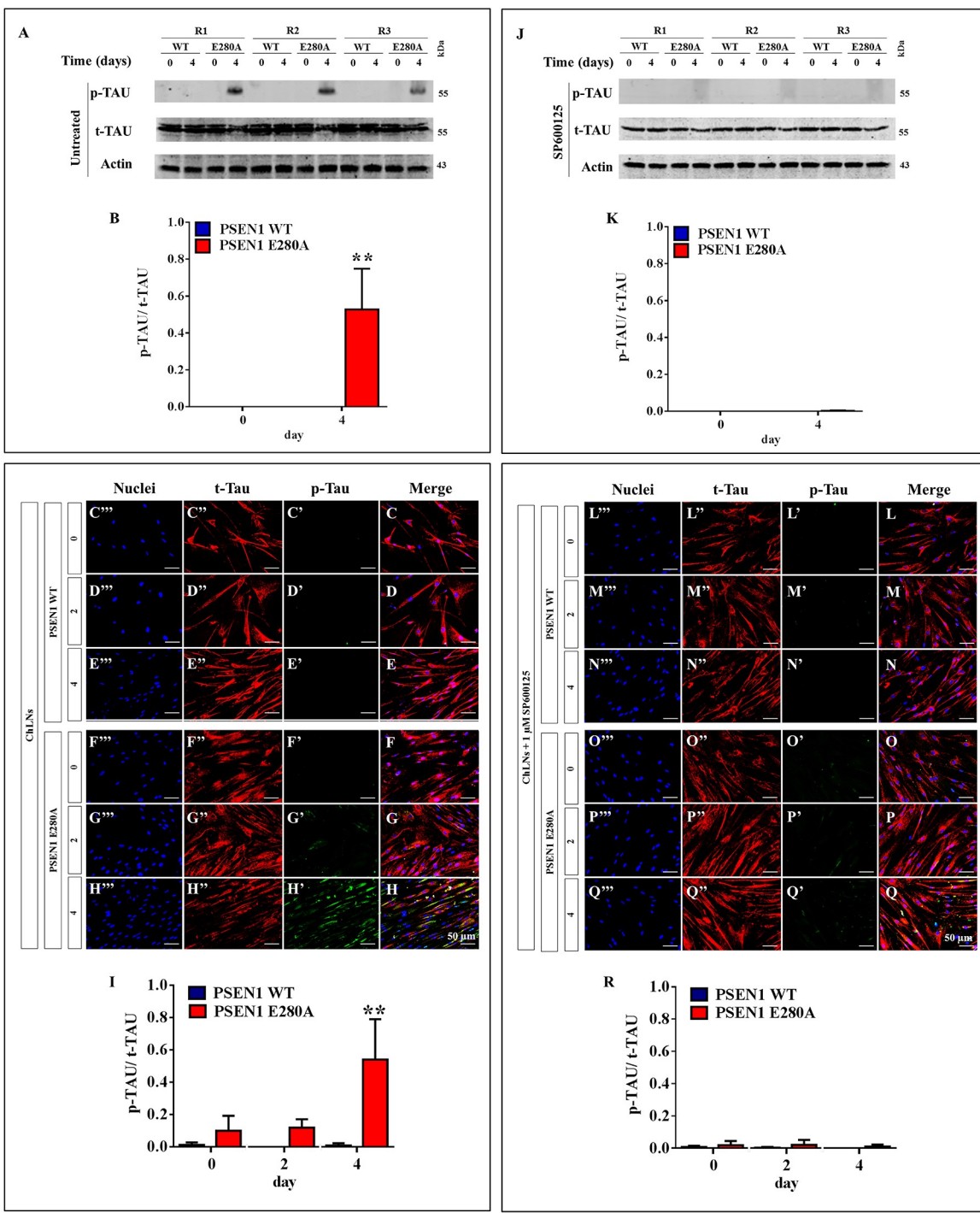

**Fig 7. PSEN1 E280A ChLNs induce phosphorylation of TAU protein.** After 7 days of transdifferentiation, WT PSEN1 and PSEN1 E280A ChLNs were left untreated **(A-I)** or treated with 1 μM JNK inhibitor SP600125 **(J-R)** in regular culture medium for 0, 2 and 4 days as indicated in the figure. After this time, the proteins in the extracts were blotted with primary antibodies against phosphorylated TAU (p-TAU), total TAU (t-TAU) and actin proteins. The intensities of the western blot bands shown in **(A or J)** were measured **(B or K)** by an infrared imaging system (Odyssey, LI-COR), and the p-TAU/t-TAU ratio was normalized to that of actin. Additionally, ChLNs were double stained as indicated in the figure **(C-H or L-Q)** with primary antibodies against p-TAU (*green*; **C'-H' or L'-Q'**) and t-TAU (*red*; **C''-H'' or L''-Q''**). The nuclei were stained with Hoechst 33342 (*blue*; **C'''-H''' or L'''-Q'''**). **(I and R)** Quantification of the p-TAU/ t-TAU fluorescence ratio. Data are expressed as the mean ± SD; *$p<0.05$; **$p<0.01$; ***$p<0.001$. The blots and figures represent 1 out of 3 independent experiments. Image magnification, 200x.

However, experimental therapies targeting Aβ have thus far been unsuccessful [50]. Several factors have probably contributed to the failures in AD drug development, including unsuitable preclinical research models that do not fully recapitulate the human disease; consequently, druggable targets remain missing [51]. Furthermore, the mechanism(s) by which hAPP/ Aβ42 might induce toxicity is not yet fully established. Because UC-MSCs possess plasticity properties enabling them to transdifferentiate into non-mesenchymal lineages, they provide a unique opportunity to study the effect of *PSEN1* mutations in neuronal cells. Most importantly, MSC-derived neurons are natural, non-genetically modified cells of the FAD PSEN1 mutation. Here, we used UC-MSCs bearing the mutation PSEN1 E280A for the first time and found that the mutation alters none of the typical MSC characteristics, such as colony-forming capacity, fibroblast-like morphology, immunophenotype and/or ability to differentiate or transdifferentiate into ChLNs. Furthermore, we did not detect any APP/ Aβ$_{42}$ produced by the cells or any other cellular alterations (e.g., ROS production, mitochondria depolarization, and oxidation of stress sensor DJ-1) in mutant MSCs. These observations suggest that the PSEN1 E280A mutation might not affect the physiology of the multipotent MS cells at this stage of development.

In this study, we report for the first time that ChLNs derived from hMSCs carrying the PSEN1 E280A mutation recapitulate typical pathologic features of AD at 4 days post transdifferentiation, i.e., eleven days of culture (7 days in Ch-N-Rm and 4 days in RCm) including increased secretion of Aβ$_{42}$, intracellular accumulation of sAPPβf, and TAU phosphorylation. Furthermore, PSEN1 E280A ChLNs not only show OS and apoptosis markers but also display Ca$^{2+}$ flux dysregulation and altered acetylcholinesterase activity. Our data provide evidence that intracellular sAPPβf specifically induces neurotoxicity through a temporal and sequential order of molecular events in PSEN1 E280A ChLNs. In agreement with commercial laboratory Western blot data (https://www.biolegend.com/en-us/products/anti-beta-amyloid-1-16-antibody-10998), the antibody 6E10 recognized monomers (4 kDa), and aggregated (~100 kDa) synthetic Aβ$_{42}$, and high molecular weight APP/ Aβ$_{42}$ (~100 kDa) in PSEN1 E280A ChLNs. Interestingly, no Aβ$_{42}$ monomers were detected in mutant ChLNs. These observations suggested aggregation of APP. In agreement with this assumption, mass spectrometry (MS) analysis identified several APP fragments (e.g., APP714, APP733, APP751, APP752, collectively named sAPPβf) but no Aβ$_{42}$ fragment was present at the gel region of HMW proteins (60–120 kDa). Outstandingly, the sAPP fragments can be generated not only by the β-secretase but also by the δ-secretase, η-secretase or Meprin β [4]. Taken together these observations suggest that sAPP fragments rather than Aβ$_{42}$ are accumulating early in the PSEN1 E280A cholinergic neurons. Accordingly, we found intracellular sAPPβf accumulation in ChLNs after 7 days of transdifferentiation (day 0), at which time fully developed ChLNs were obtained [29]. Whether intracellular APP fragments build up because full length APP processing is altered and consequently some APP fragments remain intracellular leading to cell death require further investigation. However, recent data support this view. Indeed, it has been shown that hAPP overexpression causes Aβ-independent neuronal death in olfactory sensory neurons via intrinsic apoptosis pathway [5, 6]. In contrast, others have found that the PSEN1 L166P and G384A mutations cause relocalization of γ-secretase, which significantly promotes the generation of intracellular long Aβ$_{42}$ [52]. Although it is not yet known whether the PSEN1 E280A mutation induces relocalization of γ-secretase, similar to PSEN1 mutations, our data suggest that the PSEN1 E280A mutation strongly enhances intraneuronal sAPPβf aggregates in ChLNs. These results support the view that intracellular accumulation of sAPPβf is the earliest event in the development of the neuropathological changes of AD [53]. Since the PSEN1 E280A mutation might represent a typical trans-dominant negative mutation on γ-secretase [15, 16], we do not discard the possibility that APP-derived intracellular Aβ$_{42}$ aggregates might

be a late event in the process of neurodegeneration [7, 54, 55]. Interestingly, we simultaneously found a significant increase in DCF fluorescent-positive cells, DJ-1 $Cys^{106}$–sulfonate (DJ-1 $Cys^{106}SO_3$), which is the most sensitive thiol group towards $H_2O_2$ reactivity [56], and activation of transcription factor c-JUN in PSEN1 E280A ChLNs. These observations suggest that sAPPβf (aggregates) > $H_2O_2$ > oxDJ-1$Cys^{106}$-$SO_3$ and c-JUN are the earliest events detectable in PSEN1 E280A ChLNs. These findings imply that hAPP and/or sAPPβf (high molecular weight aggregates) can trigger events related to oxidative stress and cell death [5, 6]. How then does intracellular sAPPβf generate $H_2O_2$? Several lines of evidence for the intracellular toxicity of APP have been suggested, including inhibition of mitochondrial import channels (e.g., TOM40 and TIM23), impairment of mitochondrial transport (e.g., Cytochrome c oxidase), and disruption the electron transfer [57]. Whatever the mechanism, we demonstrated for the first time that sAPPβf endogenously produces $H_2O_2$ in PSEN1 E280A ChLNs. Because $H_2O_2$ can function as a second messenger [58], it might also activate other redox proteins, such as apoptosis-signal regulating kinase 1 (ASK-1, [59]), which in turn directly or indirectly activate other signaling pathways, e.g., JNK/c-JUN [60]. We found that sAPPβf induced phosphorylation of c-JUN [61], p53 and PUMA in PSEN1 E280A ChLNs. Notably, c-JUN- and p53-dependent apoptosis is triggered by transactivation of the pro-apoptotic gene PUMA [62–64]. Remarkably, JNK can also stabilize and activate p53 [65]. Taken together, these results suggest that sAPPβf > $H_2O_2$ activates a cascade of events leading to the JNK> c-JUN, p53 > PUMA pathway. Although PUMA has been shown to cooperate with direct activator proteins (e.g., BAX) to promote mitochondrial outer membrane permeabilization (MOMP) and apoptosis [66], the exact mechanism by which MOMP occurs is not fully understood [67]. Interestingly, PSEN1 E280A ChLNs showed loss of $\Delta\Psi_m$ concomitant with overexpression of CASPASE-3 and fragmentation of nuclei at day 4 post transdifferentiation. These observations suggest that mitochondria play an important role in intracellular sAPPβf-induced apoptosis in mutant ChLNs. However, whether $\Delta\Psi_m$ dysfunction is a consequence of a direct effect of intracellular sAPPβf on the organelle or whether the damage is the result of the impact of PUMA on mitochondria is an unresolved issue. Our findings suggest that both sAPPβf / PUMA might separately or jointly damage $\Delta\Psi_m$. Taken together, these data indicate that sAPPβf induce a cascade of events in PSEN1 E280A ChLNs through the $H_2O_2$ signaling pathway involving CASP-3, as an end executer protein, and DNA fragmentation of nuclei, all indicative of apoptosis. Taken together, these data comply with the idea that intraneuronal accumulation of the APP fragments (sAPPβf) are the first step of a lethal cascade in FAD neurons.

Neuronal calcium ($Ca^{2+}$) dyshomeostasis has been proposed to play a crucial role in AD disease progression [68]. However, the mechanisms of $Ca^{2+}$ dysregulation are not clear. In contrast to Demuro and Parker [69], who found that intracellular Aβ42 oligomers disrupted cellular $Ca^{2+}$ regulation, we observed no $Ca^{2+}$ dysregulation in PSEN1 E280A ChLNs evaluated at 0, 2, and 4 days post transdifferentiation. This discrepancy can be explained by differences in experimental methodology. Whereas those authors elucidated the actions of intracellular Aβ42 by imaging $Ca^{2+}$ responses to injections of Aβ42 oligomers into *Xenopus* oocytes, we directly imaged $Ca^{2+}$ responses to endogenously generated intracellular APP in mutant ChLNs. Under the present experimental conditions, we concluded that intracellular sAPPβf did not affect $Ca^{2+}$ flux in PSEN1 E280A ChLNs. However, increasing evidence has shown that extracellular Aβ specifically interacts with nAChRs, resulting in $Ca^{2+}$ dysregulation [70]. We found that the PSEN1 E280A ChLN response to ACh was significantly reduced by day 4 post transdifferentiation. Notably, Aβ has been shown to directly affect α7 nicotinic ACh receptor (α7 nAChR) function by acting as an agonist (~100 nM) and a negative modulator (at high concentrations) [71]. Consistent with this view, we confirmed that PSEN1 E280A ChLNs secreted aberrant amounts of Aβ42 (e.g., ~2500-f.i.) compared to WT PSEN1 ChLNs. These

observations confirm that overproduction of extracellular Aβ42 is a paramount feature of the majority of PSEN1 mutations *in vitro* and *in vivo* [9], including the E280A mutation [25]. Despite these observations, further investigation is required to determine whether α7 nAChRs specifically are affected by $Aβ_{42}$ in PSEN1 E280A ChLNs. In contrast to others (e.g., [72]), our observations do not support the common view that extracellular Aβ is capable of increasing neuronal $Ca^{2+}$ flux through Aβ42-forming pores. However, we do not discard the possibility that given a longer incubation time, Aβ42 could affect $Ca^{2+}$ flux via Aβ42-forming pores in PSEN1 E280A ChLNs. Taken together, and our data suggest that extracellular Aβ42 might bind to nAChRs in PSEN1 E280A ChLNs, affecting neuronal $Ca^{2+}$ flux.

The accumulation of hyperphosphorylated TAU in neurons leads to neurofibrillary degeneration in AD [73]. Mounting evidence suggests that TAU pathogenesis is promoted by Aβ42 [74,75]. We found that PSEN1 E280A ChLNs showed hyperphosphorylation of TAU protein. In fact, PSEN1 E280A ChLNs exhibited a significantly higher p-TAU/ t-TAU ratio than WT PSEN1 ChLNs at 4 days post transdifferentiation. Because absence of Aβ42, these observations imply that sAPPβf rather than Aβ42 signaling precedes TAU phosphorylation [75, 76] in PSEN1 E280A ChLNs. These findings suggest that sAPPβf accumulation might affect TAU pathology. However, the molecular link between sAPPβf and TAU is still not yet completely defined. In agreement with others [77], our data suggest that JNK is a strong candidate TAU kinase involved in the hyperphosphorylation of TAU in PSEN1 E280A ChLNs. This assumption is supported by two observations. First, JNK phosphorylates TAU at $Ser^{202}$/ $Thr^{205}$ [47], two phosphorylation epitopes identified in the present study. Second, PSEN1 E280A ChLNs exposed to the JNK inhibitor SP600125 significantly reduced TAU phosphorylation. Given that JNK plays a pivotal role in both OS-induced apoptosis and TAU phosphorylation, these findings identify JNK as a potential therapeutic target [78]. Although we do not discard the possibility that other kinases might be implicated in TAU pathology (e.g., LRRK2, GSK-3, Cdk5), our findings suggest that JNK plays a key role in TAU hyperphosphorylation in PSEN1 E280A ChLNs. Our results suggest that PSEN1 E280A-induced neural alterations may precede Aβ42 deposition and that those alterations represent longstanding effects of intracellular sAPPβf toxicity and possibly even developmental changes. The molecular alterations might start when neurons develop into neuron-specific cholinergic-type cells or may even exist at birth. These findings may explain why functional and structural brain changes manifest in children (9–17 years old) and young individuals (18–26 years old) who are carriers of the PSEN1 E280A mutation [79, 80]. Furthermore, these observations suggest that intracellular sAPPβf toxicity is an early and slowly progressive process that might damage neuronal cells in a TAU-dependent and independent fashion (OS, $ΔΨ_m$ shutdown, apoptosis and intraneuronal $Ca^{2+}$ dysregulation) more than two decades before the stage of dementia [10, 81].

## Conclusion

We demonstrate that intracellular accumulation of sAPPβf, generation of $H_2O_2$, oxidation of the DJ-1 protein (DJ-1 $Cys^{106}$-$SO_3$), and activation of the pro-apoptosis protein c-JUN were the earliest cellular changes in PSEN1 E280A ChLNs (obtained after 7 days in Ch-N-Rm (i.e., day 0 in RCm), Fig 8 **step 1, s2, s3, s4/s5)**. These changes were followed by the activation of pro-apoptosis proteins p53 (**s6**) and PUMA (**s7**), loss of mitochondria membrane potential ($ΔΨ_m$, **s8**), activation of CASP-3 (**s9**), fragmentation of nuclei (**s10**), and complete expression of markers of apoptosis (at day 4 post transdifferentiation in RCm). These biochemical abnormalities were found concomitant with irregular secretion of Aβ42 (day 4, **s11)**, $Ca^{2+}$ flux dysregulation (**s12)**, diminished secretion of AChE (**s13)**, and hyperphosphorylation of TAU protein (day 4, **s14)**. Therefore, our data support the view that FAD PSEN1 E280A cholinergic

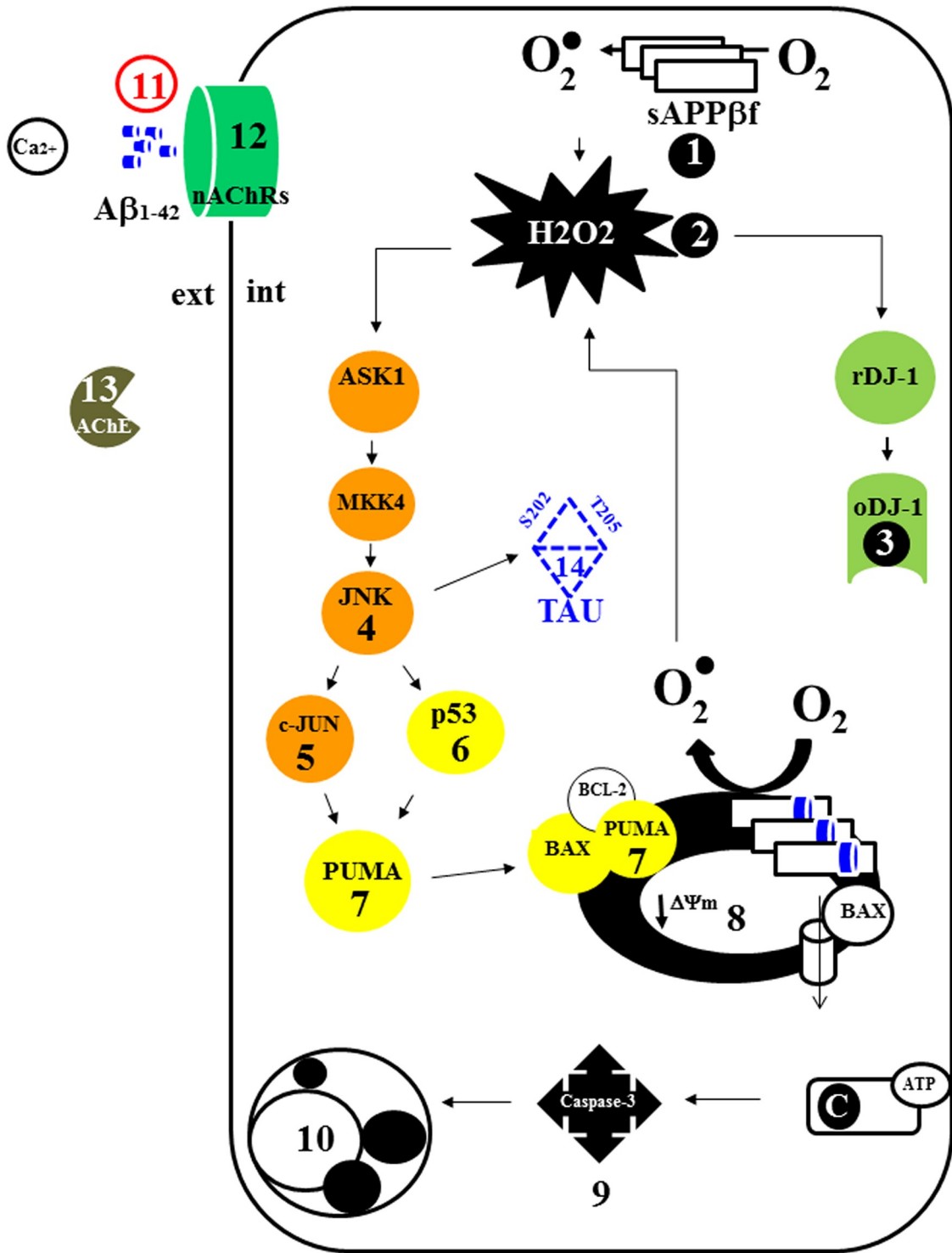

**Fig 8. Schematic effects of the sAPPβf and Aβ42 peptide in mutant PSEN1 E280A ChLNs.**

neuronal pathology is instigated by early intracellular accumulation of sAPPβf [55, 82]. These findings raise the question of whether strategies to remove extracellular Aβ$_{42}$ such as immuno-therapy [83] should be complemented with treatments to remove intracellular sAPPβf [84] and OS (i.e., H$_2$O$_2$) to avoid apoptosis and TAU pathology and treatment with nAChR

agonists to increase neuronal functionality. The present findings naturally (i.e., not genetically manipulated) recapitulated for the first time the neuropathological features of FAD PSEN1 E280A. We anticipate that the present *in vitro* model will inspire new and innovative therapies for early onset PSEN1 E280A patients. Despite the fact that the mutation PSEN1 E280A is 100% penetrant in those individuals bearing such mutation, and given that our findings are based on ChLNs obtained from a single PSEN1 E280A MSCs, the conclusions from such analyses require verification through additional studies with larger sample size.

## Supporting information

**S1 Fig. Identification of PSEN1 genotype in WT PSEN1 and PSEN1 E280A in WJ-MSCs.** The PSEN1 E280A mutation was detected as described in *Materials and Methods* section. According to different mobility electrophoretic patterns, samples were classified as wild-type (WT) or mutant PSEN1 E280A when compared to PSEN1 E280A carrier (positive case NB code#18233) or wild type PSEN1 genotype (NB code#18574). ***Abbreviations***: TBC# = tissue bank code number; F = female, M = male; (+) = positive; (-) = negative; numbers on the left are molecular size markers expressed in bp.
(TIF)

**S2 Fig. Apolipoprotein E (APOE) genotype in WT PSEN1 and PSEN1 E280A (n = 1) WJ-MSCs.** Electrophoretic separation of *HhaI* fragments after gene amplification of DNA from WT PSEN1 and PSEN1 E280A WJ-MSCs. The DNA band pattern was determined according to ref. [29]. Numbers on the left are molecular size markers expressed in bp. ***Abbreviations*: TBC# = tissue bank code number; F = female, M = male**.
(TIF)

**S3 Fig. WT PSEN1 and PSEN1 E280A WJ-MSCs show similar levels of intracellular APP/ Aβ$_{42}$ and oxidized DJ-1.** WT PSEN1 and PSEN1 E280A WJ-MSCs were cultured in MCm for 7 days, and then cultured for 4 additional days in regular culture medium (RCm). Then, the proteins in the extracts and control extracts were blotted with primary antibodies against Aβ$_{42}$, oxDJ-1Cys$^{106}$ and actin proteins. The intensities of the western blot bands shown in (**A**) were measured (**B, C**) by an infrared imaging system (Odyssey, LI-COR), and the intensity was normalized to that of actin. Control lysates were included to validate the results. Additionally, after 0, 2 and 4 days, WJ-MSCs were double stained as indicated in the figure (**D-I**) with primary antibodies against Aβ$_{42}$ (*red*; **D'-I'**) and oxDJ-1Cys$^{106}$ (*green*; **D'-I"**). The nuclei were stained with Hoechst 33342 (**blue; D"-I"'**). (**J**) Quantification of Aβ$_{42}$ fluorescence intensity. (**K**) Quantification of oxDJ-1Cys$^{106}$ fluorescence as MFI. Data are expressed as the mean ± SD; *$p < 0.05$; **$p < 0.01$; ***$p < 0.001$. The blots and figures represent 1 out of 3 independent experiments. Image magnification, 200x.
(TIF)

**S4 Fig. WT PSEN1 and PSEN1 E280A WJ-MSCs show similar levels of mitochondrial membrane potential (ΔΨ$_m$), intracellular reactive oxygen species (ROS) and DNA fragmentation.** Representative histograms showing DiOC$_6$(3)$^{low}$ (**A**), DCF+ (**B**) and SubG$_1$ (**C**) populations from WT PSEN1 (blue) and PSEN1 E280A (Red) WJ-MSCs after 7 days in MCm plus 4 days in RCm. Representative MitoTracker (**D'-I'**), DCF (**D"-I"**), Hoechst (**D"'-I"'**) and merged (**D-I**) pictures of WT PSEN1 and PSEN1 E280A WJ-MSCs after 0, 2 and 4 days in RCm. (**J**) Quantification of MitoTracker fluorescence intensity. (**K**) Quantification of DCF fluorescence intensity. Data are expressed as the mean ± SD; *$p < 0.05$; **$p < 0.01$; ***$p < 0.001$. The histograms and figures represent 1 out of 3 independent experiments. Image

magnification, 200x.
(TIF)

**S5 Fig.** Mascot Search Results of (**A**) MS/MS fragmentation of LPTTAASTPDAVDK, and (**B**) MS/MS fragmentation of AVIQHFQEK. (**C**) List of proteins identified in the PAGE gel region of high molecular weight between 60 and 125 kDa.
(PDF)

**S1 File.**
(ZIP)

**S1 Data.**
(XLSX)

## Acknowledgments

We greatly acknowledge KA Herrera-Galeano (Nurse), F Piedrahita (Nurse), L Lopez (Psychologist) and AA Espinosa-Rojas (MD) for technical support for obtaining human umbilical cord tissue at Hospital San Juan de Dios, Yarumal, Colombia. We specially acknowledge the newborns' parents for human umbilical cord tissue. We thank Dr. A Shevchenko (Head of Mass Spectrometry, Max Plank Institute of Molecular Cell Biology and Genetics, Dresden-Germany) for LC-MS/MS analysis.

## Author Contributions

**Conceptualization:** Carlos Velez-Pardo, Marlene Jimenez-Del-Rio.

**Data curation:** Viviana Soto-Mercado, Miguel Mendivil-Perez, Carlos Velez-Pardo, Marlene Jimenez-Del-Rio.

**Formal analysis:** Viviana Soto-Mercado, Miguel Mendivil-Perez, Carlos Velez-Pardo.

**Funding acquisition:** Francisco Lopera, Marlene Jimenez-Del-Rio.

**Investigation:** Viviana Soto-Mercado, Miguel Mendivil-Perez.

**Methodology:** Viviana Soto-Mercado, Miguel Mendivil-Perez.

**Project administration:** Marlene Jimenez-Del-Rio.

**Resources:** Francisco Lopera, Marlene Jimenez-Del-Rio.

**Supervision:** Carlos Velez-Pardo, Francisco Lopera.

**Writing – original draft:** Carlos Velez-Pardo, Marlene Jimenez-Del-Rio.

**Writing – review & editing:** Viviana Soto-Mercado, Miguel Mendivil-Perez, Carlos Velez-Pardo, Francisco Lopera, Marlene Jimenez-Del-Rio.

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
