## [Decision Letter · Decision Letter 0]

16 Sep 2019

PONE-D-19-22448

Cholinergic-like neurons carrying PSEN1 E280A mutation from familial Alzheimer’s disease reveal intraneuronal Abeta42 accumulation, hyperphosphorylation of TAU, oxidative stress, apoptosis and Ca2+ flux dysregulation: Therapeutic implications

PLOS ONE

Dear Dr. Jimenez-Del-Rio,

Thank you for submitting your manuscript to PLOS ONE. After careful consideration, we feel that it has merit but does not fully meet PLOS ONE’s publication criteria as it currently stands. Therefore, we invite you to submit a revised version of the manuscript that addresses the points raised during the review process.

We would appreciate receiving your revised manuscript by Oct 31 2019 11:59PM. To enhance the reproducibility of your results, we recommend that if applicable you deposit your laboratory protocols in protocols.io, where a protocol can be assigned its own identifier (DOI) such that it can be cited independently in the future. For instructions see: http://journals.plos.org/plosone/s/submission-guidelines#loc-laboratory-protocols

We look forward to receiving your revised manuscript.

Kind regards,

Firas H Kobeissy, PhD

Academic Editor

PLOS ONE

Journal Requirements:

2. Thank you for letting us know you require assistance with making your Data publicly available.

Please note our data availability policy requires authors to make all data underlying the findings described in their manuscript fully available without restriction at the time of publication. When specific legal or ethical requirements prohibit public sharing of a dataset, authors must indicate how researchers may obtain access to the data. (https://journals.plos.org/plosone/s/data-availability)

If the data is not restricted for public sharing, we strongly recommend depositing your data in a public repository. Please see our Data availability policy for a list of recommended repositories (https://journals.plos.org/plosone/s/data-availability#loc-recommended-repositories). In addition, it is also acceptable to upload the data as a Supporting Information file.

If the data is restricted from being shared public for ethical or legal reasons, please make the data available upon request and identify the group to which requests should be submitted (e.g., a named data access committee or named ethics committee). The reasons for restrictions on public data deposition must also be specified. Note that it is not acceptable for an author to be the sole named individual responsible for ensuring data access. (https://journals.plos.org/plosone/s/data-availability#loc-acceptable-data-sharing-methods)

If you have any questions or concerns please let us know and we would be happy to provide further assistance.

Additional Editor Comments (if provided):

Dear Dr. Jimenez-Del-Rio,

thank you for your submission, the work presented was well-received and evaluated by top experts in the area of Alzheimer’s disease.

there are some concerns related to the selection of the 10 different umbilical cord samples owith one of them carrier of the PSEN1 E280A mutation from which one was selected for comparison with the E280A mutation.

I would request that the authors perform a mass spec analysis of the proposed oligomeric Abeta 42 in Figure 3A (the band should provide some peptides of Abeta proteins).

Reviewer # 2 has excellent comments that need to be addressed

Thank you Firas Kobeissy

Reviewers' comments:

Reviewer's Responses to Questions

**Comments to the Author**

1. Is the manuscript technically sound, and do the data support the conclusions?

Reviewer #1: Yes

Reviewer #2: Partly

2. Has the statistical analysis been performed appropriately and rigorously? 

Reviewer #1: I Don't Know

Reviewer #2: Yes

3. Have the authors made all data underlying the findings in their manuscript fully available?

Reviewer #1: No

Reviewer #2: Yes

4. Is the manuscript presented in an intelligible fashion and written in standard English?

Reviewer #1: Yes

Reviewer #2: Yes

5. Review Comments to the Author

Reviewer #1: This manuscript described a new and effective way of cell culture model that provides novel opportunities to study the mechanisms of AD on cellular level, which potentially will become a good model for therapeutic study. In the future, more studies should be done using this model to compare different mutations. Better including siblings as controls to limit the genetic background variation. If this method is widely adopted, a large MSCs bank should be setup for the study of all kinds of human diseases. The data is very impressive. There are a few small things I would like to address:

1. You collected 10 samples. Only 1 of them has the mutation. You also only randomly picked 1 out of 9 as the WT control. Did you ever test any of the other 8? If yes, could you show the data? Any of the other 8 have ApoE 3/4, which could be a better control. In fact, you should use more than one WT control, since you have enough resource. If possible, could you add another one, preferred ApoE 3/4, to the data?

2. Since you only tripicate of the experiment, could you run all the 3 repeats on the same gels? Please use that the replace the western blot pictures on figure 2, 3 and 5.

3. In figure 3, we can see pretty high levels of Abeta42 on Figure A, but almost 0 on figure J. How to explain it?

4. The Figure 3 E and F showed nothing, comparing to H and I, which is not consistent with western blot at figure A.

5. At figure A, you didn't detect any monomer of Abeta42 at all, which is not consistent from our experience. In order to test whether your anitbody is specific enough for Abeta, could you repeat the same figure using another Abeta or APP antibody, such as 6E10?

6. The quality of the western blot figure of Figure 3A on OxDJ-1 Cys106 is too poor. Do you have a better picture? Anyway, you should rerun the whole gel with all 3 repeat on the same gel. You can buy 18 well gel from Bio-Rad.

7. You talked about Ca2+ flux at many places in the manuscript, but there is no data to show it. Did you forget to put the figures in?

8. At the method part of the Isolation and Expansion of hWJ-MSCs, 2000 rpm, using x g is better, since people will have different centrifuge than yours.

9. At the method part of western blot, could you list the dilution of the primary antibodies so that other labs can follow?

10. On page 16, did you use 400x objective of the microscope. Usually 100x is the highest amplification of the objective.

Reviewer #2: The paper by Soto-Mercado examines the production and oligomerization of Abeta in induced cholinergiec neurons derived from umbilical cord meschymal stem cells. The authors report that they initially began the study with 10 different umbilical cord samples of which only 1 was a gene carrier for the PSEN1 E280A mutation. The study then focuses on cultures derived from one of the 9 samples lacking a mutation for comparison to the one culture with the E280A mutation. A brief explanation on how the one control sample was chosen would be useful. Was it chosen at random, or for some other propery?. It is a weakness of the study that other cultures from non-mutant carriers were not characterized to provide a view of the variability of the characteristics of these cultures. As such, the study is really a case-stud of one isolated MSC culture derived from one individual compare to a single individual without the mutation. In my opinion, the conclusions that the authors draw from the study of these cultures are over-stated.

The authors state that the major implication of their study is that they find an important role for intracellular Abeta in cholinergic neurons expressing mutant SOD1. Given the emphasis on this conclusion, the authors need to provide more evidence that the bands that are detected as oligomeric Abeta 42 in Figure 3A are indeed Abeta. Although the antibody appears to be specific to the C-terminus of Abeta42, the authors need to provide additional evidence that the immunoreactive bands in Fig. 3A are really oligomeric Abeta 42. For example, can the authors demonstrate that the same bands are recognized by an antibody that recognizes the N-terminus of Abeta such as clone 1E8 (MABN639 from Millipore/Sigma). Additionally, given the intensity of antibody immunoreactivity in western blots of WT cells at 4 hours, it is surprising that the immunostaining of these cells in panels D-F is so weak. I would again like to see staining with an N-terminal antibody Abeta antibody.

6. PLOS authors have the option to publish the peer review history of their article (what does this mean?). If published, this will include your full peer review and any attached files.

Reviewer #1: No

Reviewer #2: No

---

## [Author Response · Author response to Decision Letter 0]

25 Feb 2020

We are grateful to editor and reviewers for the time spent and pertinent suggestions intended to improve our manuscript. The response to the specific comments is listed below. 

Journal Requirements: 

http://www .journals.plos.org/plosone/s/file?id=wjVg/PLOSOne_formatting_sample_main_body.pdfand

http://www .journals.plos.org/plosone/s/file?id=ba62/PLOSOne_formatting_sample_title_authors_affiliations.pdf

Response (R:/) We have submitted our manuscript as required by PLOS One journal. Please let us know if something is missing to promptly respond. 

2. Thank you for letting us know you require assistance with making your Data publicly available. Please note our data availability policy requires authors to make all data underlying the findings described in their manuscript fully available without restriction at the time of publication. When specific legal or ethical requirements prohibit public sharing of a dataset, authors must indicate how researchers may obtain access to the data. (https://journals.plos.org/plosone/s/data-availability). 

(R:/) The following statement is now included in the manuscript. In addition we are submitting Supporting information figures (blots/ gels) and supporting information data (Excel format). 

Data availability 

Anonymized clinical and genetic data are available upon request, subject to an internal review by F.L. and MJDelR to ensure that the participants’ anonymity, confidentiality, and PSEN1 E280A carrier or non-carrier status are protected, completion of a data sharing agreement, and in accordance with the University of Antioquia’s review board and institutional guidelines. Experimental data are available upon request, subject to University of Antioquia guidelines. Material requests and data requests will be considered based on a proposal review, completion of a material transfer agreement (MTA) and/or a data use agreement, and in accordance with the University of Antioquia guidelines. Please submit requests for participant-related clinical and genetic data to FL (francisco.lopera@gna.org.co) and requests for experimental data to VS-M, MM-P, CV-P (vivianamarcelasoto@gmail.com; miguelangelmendivil@gmail.com; calberto.velez@udea.edu.co).

If the data is not restricted for public sharing, we strongly recommend depositing your data in a public repository. Please see our Data availability policy for a list of recommended repositories (https://journals.plos.org/plosone/s/data-availability#loc-recommended-repositories). In addition, it is also acceptable to upload the data as a Supporting Information file. If the data is restricted from being shared public for ethical or legal reasons, please make the data available upon request and identify the group to which requests should be submitted (e.g., a named data access committee or named ethics committee). The reasons for restrictions on public data deposition must also be specified. Note that it is not acceptable for an author to be the sole named individual responsible for ensuring data access. https://journals.plos.org/plosone/s/dataavailability#loc-acceptable-data-sharing-methods)

3. PLOS ONE now requires that authors provide the original uncropped and unadjusted images underlying all blot or gel results reported in a submission’s figures or Supporting Information files. This policy and the journal’s other requirements for blot/gel reporting and figure preparation are described in detail at https://journals.plos.org/plosone/s/figures#loc-blotand-gel-reporting-requirements and https://journals.plos.org/plosone/s/figures#loc-preparing-figures-from-image-files.

When you submit your revised manuscript, please ensure that your figures adhere fully to these guidelines and provide the original underlying images for all blot or gel data reported in your submission. See the following link for instructions on providing the original image data: https://journals.plos.org/plosone/s/figures#loc-original-images-for-blots-and-gels. In your cover letter, please note whether your blot/gel image data are in Supporting Information or posted at a public data repository, provide the repository URL if relevant, and provide specific details as to which raw blot/gel images, if any, are not available. Email us at plosone@plos.orgif you have any questions.

(R:/) We are providing the original uncropped and unadjusted images underlying all blots or gels reported as Supporting Information and Supporting Information data (in Excel format. 

4. We note that you have included the phrase “data not shown” in your manuscript. Unfortunately, this does not meet our data sharing requirements. PLOS does not permit references to inaccessible data. We require that authors provide all relevant data within the paper, Supporting Information files, or in an acceptable, public repository. Please add a citation to support this phrase or upload the data that corresponds with these findings to a stable repository (such as Figs hare or Dryad) and provide and URLs, DOIs, or accession numbers that may be used to access these data. Or, if the data are not a core part of the research being presented in your study, we ask that you remove the phrase that refers to these data.

(R:/) The phrase “data not shown” has been removed from the manuscript. 

Editor Dr. Firas Kobeissy’s Comments

I would request that the authors perform a mass spec analysis of the proposed oligomeric Abeta 42 in Figure 3A (the band should provide some peptides of Abeta proteins).

(R:/) This is a most pertinent requirement. As suggested by editor, the proposed oligomeric Abeta 42 was analyzed by LC-MS/MS (please see Supporting Material (S5A, B, and C Fig), Results and Discussion sections). Surprisingly, “To confirm the identity of the intracellular APP/ A�42 aggregates displayed in the Western blot (Fig. 3A, arrowhead), we used LC-MS/MS technique. The LC-MS/MS analysis identified two peptides 364LPTTAASTPDAVDK377 (S5A Fig) and 430AVIQHFQEK438 (S5B Fig) that showed high homology with APP fragments (e.g., APP714, APP733, APP751, APP752, S5C Fig), hereafter sAPP�f. Strikingly, no A�42 fragment was identified in the interrogated gel region”. (pag. 25, line 15-20). We therefore concluded that “Taken together these observations suggest that sAPP fragments rather than Aβ42 are accumulating early in the PSEN1 E280A cholinergic neurons” (pag. 34, line 19-21). Furthermore “Since the PSEN1 E280A mutation might represent a typical trans-dominant negative mutation on �-secretase (Heilig et al., 2013; Zhou et al., 2017), we do not discard the possibility that APP-derived intracellular Aβ42 aggregates might be a late event in the process of neurodegeneration (Takahashi et al., 2017; Wirths et al., 2004; LaFerla et al., 2007)” (pag. 35, line 13-15). 

Reviewers' comments

Reviewer's Responses to Questions

Comments to the Author

1. Is the manuscript technically sounds, and do the data support the conclusions?

Reviewer #1: Yes

Reviewer #2: Partly

2. Has the statistical analysis been performed appropriately and rigorously?

Reviewer #1: I Don't Know (R:/) On pag. 19, in original manuscript, we provided a detailed description of the statistical analysis of the data (Data analysis section) including Ref. [38]. 

Reviewer #2: Yes 

3. Have the authors made all data underlying the findings in their manuscript fully available?

Reviewer #1: No (R:/) A data availability statement is now included in the manuscript (pag 42), supporting information figures and supporting information data (in excel format) are now provided. 

Reviewer #2: Yes

 4. Is the manuscript presented in an intelligible fashion and written in standard English?

Reviewer #1: Yes

Reviewer #2: Yes

5. Review Comments to the Author

Reviewer #1: This manuscript described a new and effective way of cell culture model that provides novel opportunities to study the mechanisms of AD on cellular level, which potentially will become a good model for therapeutic study. In the future, more studies should be done using this model to compare different mutations. Better including siblings as controls to limit the genetic background variation. If this method is widely adopted, a large MSCs bank should be setup for the study of all kinds of human diseases. The data is very impressive. There are a few small things I would like to address:

1. You collected 10 samples. Only 1 of them has the mutation. You also only randomly picked 1 out of 9 as the WT control. Did you ever test any of the other 8? If yes, could you show the data? Any of the other 8 have ApoE 3/4, which could be a better control. In fact, you should use more than one WT control, since you have enough resource. If possible, could you add another one, preferred ApoE 3/4, to the data? 

(R:/) We did not test all samples but only for the selected one described in the present work (i.e. wild type and mutant PSEN1 E280A). Based on the fact that the mutation PSEN1 E280A is 100% penetrant in those individuals bearing the mutation, we just follow the following selection process. (Pag.7 lines 5-11) “The PSEN1 E280A mutation was detected by PCR using mismatch primers and digestion of the products with Bsm1 [28]. Digested products were separated on a 3% agarose gel. According to different mobility electrophoretic patterns, samples were classified as wild-type (WT) or mutant PSEN1 E280A when compared to PSEN1 E280A carrier (positive case NeuroBank (NB) code #18233) or wild type PSEN1 genotype (NB code#18574). The TBC# WJMSC-12 (female) sample was identified for PSEN1 E280A mutation (S1 Fig). For comparative purposes, we pre-selected female WT PSEN1 samples (TBC# WJMSC-11, -15, -17), and TBC# WJMSC-11 WT PSEN1 was randomly selected for further experiments”. As required by referee, genotype of 10 samples is now included (S1 Fig).

It is well known that the strongest known genetic risk factor for the development of late-onset Alzheimer disease is inheritance of the apolipoprotein (APO) E4 allele. Indeed, lipoprotein transport system is mediated by high density lipoprotein-sized. The APO E-containing lipoproteins are synthesized and secreted by glial cells (primarily astrocytes). The astrocyte-derived APO E-containing lipoproteins can bind to, and be internalized by, receptors of the low density lipoprotein receptor superfamily that are located on the surface of neurons. We therefore think that the APOE profile of the wild type APOE 3/3 and APOE 3/4 ChLNs are not critical for the analysis of early onset FAD cholinergic neurons as shown in the present investigation. We consider that the APOE might be informative for future studies where neurons and astrocytes interaction would be investigated. The APOE profile of samples (n=10) are now included (S2 Fig).

2. Since you only tripicate of the experiment, could you run all the 3 repeats on the same gels? Please use that the replace the western blot pictures on figure 2, 3 and 5.

(R:/) As suggested by the reviewer, all the 3 repeats of the experiments are now run on the same gels on Fig 2, 3 and 5. Please note that we now included Fig 7 as for figures 2,3,5. 

3. In figure 3, we can see pretty high levels of Abeta42 on Figure A, but almost 0 on figure J. How to explain it?

(R:/) The referee remark is right. We think the difference between Fig 3A and Fig J is technical. While Western blot (WB) uses cell extract for protein analysis, IF analysis of protein is at single-cell level. Therefore, Western blot is more sensible than IF. Alternatively, the antibody which recognized c-terminal Abeta (AB5078P) may not be optimal. To get solved this problem, as suggested by referee 1 and 2, we have used the antibody 6E10. Noticeably, the use of this antibody now corrects for this. The results are now presented in Fig. 3A and 3J with similar observations for both WB and IF. However, as LC-MS/MS analysis proved, no A�42 but rather sAPP� fragments were identified in the WB-HMW band.

4. The Figure 3E and F showed nothing, comparing to H and I, which is not consistent with western blot at figure A.

(R:/) As noted in item 3, this discrepancy has been solved with the use of antibody 6E10. 

5. At figure A, you didn't detect any monomer of Abeta42 at all, which is not consistent from our experience. In order to test whether your anitbody is specific enough for Abeta, could you repeat the same figure using another Abeta or APP antibody, such as 6E10?

(R:/) As suggested by the referee, we have repeated the detection of synthetic Abeta 42 with antibody 6E10. As shown in blot, and similar to c-terminus Abeta antibody AB5078P, no Abeta42 monomers were detected. 

6. The quality of the western blot figure of Figure 3A on OxDJ-1 Cys106 is too poor. Do you have a better picture? Anyway, you should rerun the whole gel with all 3 repeat on the same gel. You can buy 18 well gel from Bio-Rad.

(R:/) As suggested by the reviewer, the quality of the WB has been improved. We have rerun the whole gel with all 3 repeat on the same gel. Please note however that since there were no differences between day 2 and 4, we selected 0 and 4 day to rerun the gels. Also note that 15 well gel but not 18 well gel was selected for technical easiness and availability of the complete set of apparatus involving protein electrophoresis apparatus plus + Western blotting apparatus, among other reagents. Please note that similar data at day 0 and 4 were obtained as submitted originally. 

7. You talked about Ca2+ flux at many places in the manuscript, but there is no data to show it. Did you forget to put the figures in?

(R:/) Fig 6 was successfully submitted in the original ms wherein the Ca2+ flux experiments were shown. Please see it again in the present ms version. 

8. At the method part of the Isolation and Expansion of hWJ-MSCs, 2000 rpm, using x g is better, since people will have different centrifuge than yours.

(R:/) As suggested, rpm has been changed to x g through material and method text. 

9. At the method part of western blot, could you list the dilution of the primary antibodies so that other labs can follow?

(R:/) The dilution of the primary antibodies appeared in the original manuscript (pag. 11 and 12). 

10. On page 16, did you use 400x objective of the microscope. Usually 100x is the highest amplification of the objective. (R:/) This error has been corrected. It is 100x as noted. 

Reviewer #2: The paper by Soto-Mercado examines the production and oligomerization of Abeta in induced cholinergiec neurons derived from umbilical cord mesenchymal stem cells. The authors report that they initially began the study with 10 different umbilical cord samples of which only 1 was a gene carrier for the PSEN1 E280A mutation. The study then focuses on cultures derived from one of the 9 samples lacking a mutation for comparison to the one culture with the E280A mutation. A brief explanation on how the one control sample was chosen would be useful. Was it chosen at random, or for some other property? 

(R:/) As suggested by referee 1 and 2, a brief explanation as to how samples were chosen is now included (pag. 7, lines 5-11). Based on the fact that the mutation PSEN1 E280A is 100% penetrant in those individuals bearing the mutation, we just follow the following selection process. “The PSEN1 E280A mutation was detected by PCR using mismatch primers and digestion of the products with Bsm1 [28]. Digested products were separated on a 3% agarose gel. According to different mobility electrophoretic patterns, samples were classified as wild-type (WT) or mutant PSEN1 E280A when compared to PSEN1 E280A carrier (positive case NeuroBank (NB) code #18233) or wild type PSEN1 genotype (NB code#18574). The TBC# WJMSC-12 (female) sample was identified for PSEN1 E280A mutation (S1 Fig). For comparative purposes, we pre-selected female WT PSEN1 samples (TBC# WJMSC-11, -15, -17), and TBC# WJMSC-11 WT PSEN1 was randomly selected for further experiments”. 

It is a weakness of the study that other cultures from non-mutant carriers were not characterized to provide a view of the variability of the characteristics of these cultures. As such, the study is really a case-study of one isolated MSC culture derived from one individual compare to a single individual without the mutation. 

(R:/) The reviewer is right, this is a case-study of one isolated MSC culture derived from one individual compare to a single individual without the mutation. As mentioned before, since the mutation is 100% penetrant, we consider that one single control might suffice for the purpose of the present study. For the sake of clarity, the sentence “Despite the fact that the mutation PSEN1 E280A is 100% penetrant in those individuals bearing such mutation, and given that our findings are based on ChLNs obtained from a single PSEN1 E280A MSCs, the conclusions from such analyses require verification through additional studies with larger sample size ” is now included (pag 40, line 10-13). 

In my opinion, the conclusions that the authors draw from the study of these cultures are over-stated. 

(R:/) Given the observation that no A�42 but rather sAPP� fragments were identified in the WB-HMW band by LC-MS/MS analysis, we think that the conclusions are now adjusted to observations. Furthermore, for the sake of clarity, the sentence “Despite the fact that the mutation PSEN1 E280A is 100% penetrant in those individuals bearing such mutation, and given that our findings are based on ChLNs obtained from a single PSEN1 E280A MSCs, the conclusions from such analyses require verification through additional studies with larger sample size ” (pag. 42, line 10-13). 

The authors state that the major implication of their study is that they find an important role for intracellular Abeta in cholinergic neurons expressing mutant SOD1. (R:/) in mutant PSEN1 not in SOD1. Given the emphasis on this conclusion, the authors need to provide more evidence that the bands that are detected as oligomeric Abeta 42 in Figure 3A are indeed Abeta. (R:/) This is most pertinent remark. As suggested by the reviewer and Editor, we have now repeated all experiments with antibody 6E10, and submitted band (aggregate Abeta) to mass spectrometry analysis (please see results section). Surprisingly, “To confirm the identity of the intracellular APP/ A�42 aggregates displayed in the Western blot (Fig. 3A, arrowhead), we used LC-MS/MS technique. The LC-MS/MS analysis identified two peptides 364LPTTAASTPDAVDK377 (S5A Fig) and 430AVIQHFQEK438 (S5B Fig) that showed high homology with APP fragments (e.g., APP714, APP733, APP751, APP752, S5C Fig), hereafter sAPP�f. Strikingly, no A�42 fragment was identified in the interrogated gel region”. (pag. 25, line 15-20). We therefore concluded that “Taken together these observations suggest that sAPP fragments rather than Aβ42 are accumulating early in the PSEN1 E280A cholinergic neurons” (pag. 34, line 19-21). Furthermore “Since the PSEN1 E280A mutation might represent a typical trans-dominant negative mutation on �-secretase (Heilig et al., 2013; Zhou et al., 2017), we do not discard the possibility that APP-derived intracellular Aβ42 aggregates might be a late event in the process of neurodegeneration (Takahashi et al., 2017; Wirths et al., 2004; LaFerla et al., 2007)” (pag. 35, line 13-15).

Although the antibody appears to be specific to the C-terminus of Abeta42, the authors need to provide additional evidence that the immunoreactive bands in Fig. 3A are really oligomeric Abeta 42. For example, can the authors demonstrate that the same bands are recognized by an antibody that recognizes the N-terminus of Abeta such as clone 1E8 (MABN639 from Millipore/Sigma). 

(R:/) As suggested by the reviewer, we have now repeated all experiments with antibody 6E10 and have found consistently that N-terminus antibody (6E10) provides similar results and conclusions as antibody that recognized c-terminus of Abeta. Additionally, we included LC-MS/MS analysis. We conclude that antibody 6E10 recognized high molecular aggregates of sAPP�. 

Additionally, given the intensity of antibody immunoreactivity in western blots of WT cells at 4 hours, it is surprising that the immunostaining of these cells in panels D-F is so weak. I would again like to see staining with an N-terminal antibody Abeta antibody. 

(R:/) As mentioned, this immunostaining weakness has now been resolved with the use of (6E10). 

6. PLOS authors have the option to publish the peer review history of their article (what does this mean?). If published, this will include your full peer review and any attached files.

Do you want your identity to be public for this peer review? For information about this choice, including consent withdrawal, please see our Privacy Policy.

Reviewer #1: No

Reviewer #2: No

---

## [Decision Letter · Decision Letter 1]

27 Apr 2020

PONE-D-19-22448R1

Cholinergic-like neurons carrying PSEN1 E280A mutation from familial Alzheimer’s disease reveal intraneuronal sAPPβ fragments accumulation, hyperphosphorylation of TAU, oxidative stress, apoptosis and Ca2+ dysregulation: Therapeutic Implications

PLOS ONE

Dear Dr. Jimenez-Del-Rio,

Thank you for submitting your manuscript to PLOS ONE. After careful consideration, we feel that it has merit but does not fully meet PLOS ONE’s publication criteria as it currently stands. Therefore, we invite you to submit a revised version of the manuscript that addresses the points raised during the review process.

We would appreciate receiving your revised manuscript by Jun 11 2020 11:59PM. To enhance the reproducibility of your results, we recommend that if applicable you deposit your laboratory protocols in protocols.io, where a protocol can be assigned its own identifier (DOI) such that it can be cited independently in the future. For instructions see: http://journals.plos.org/plosone/s/submission-guidelines#loc-laboratory-protocols

We look forward to receiving your revised manuscript.

Kind regards,

Firas H Kobeissy, PhD

Academic Editor

PLOS ONE

Additional Editor Comments (if provided):

Dear Authors,

thank you for the submission and we truly appreciate the hard work in answering the previous concerns.

there are some minor comments related to the result sections and data interpretation pertaining to aim APP and Abeta 42 levels reporting in the figures.

Please address these concerns.

Thank you

Reviewers' comments:

Reviewer's Responses to Questions

**Comments to the Author**

1. If the authors have adequately addressed your comments raised in a previous round of review and you feel that this manuscript is now acceptable for publication, you may indicate that here to bypass the “Comments to the Author” section, enter your conflict of interest statement in the “Confidential to Editor” section, and submit your "Accept" recommendation.

Reviewer #1: All comments have been addressed

Reviewer #2: (No Response)

2. Is the manuscript technically sound, and do the data support the conclusions?

Reviewer #1: Yes

Reviewer #2: Partly

3. Has the statistical analysis been performed appropriately and rigorously? 

Reviewer #1: Yes

Reviewer #2: I Don't Know

4. Have the authors made all data underlying the findings in their manuscript fully available?

Reviewer #1: Yes

Reviewer #2: Yes

5. Is the manuscript presented in an intelligible fashion and written in standard English?

Reviewer #1: Yes

Reviewer #2: Yes

6. Review Comments to the Author

Reviewer #1: This manuscript described a new and effective way of cell culture model that provides novel opportunities to study the mechanisms of AD on cellular level, which potentially will become a good model for therapeutic study. The data is very impressive. The corrected version already addressed all my comments and included a lot of extra effort. It is a good study.

Reviewer #2: The authors have largely addressed my original criticisms. However, the new data raise a new question that requires clarification. The authors indicate that the induced neurons with the PS1 mutation have increased levels of an APP fragment. In Figure 3, the authors have 2 time points post differentiation – 0 day and 4 days. I really see also most no change in the levels of most of what is measured over the 4 days. In Figure S3, the authors show that MSCs in regular medium with the PS1 mutation have no detectable levels of the intracellular APP fragments. I am struggling to understand why the 0 day for the mutant cells in the differentiation paradigm do not match the undifferentiated MSCs? A similar question applies to the oxidized DJ-1 data. The only proteins showing an obvious temporal change in levels are Caspase 3 and p53. In the Discussion, the authors state that day 0 is after 7 days of differentiation – in which case day is really day 7 and day 4 is really day 11. Please be precise in your descriptions.

The abstract mentions increased secretion of Abeta 42 by the mutant cells, but I did not see any data to support this conclusion.

If the APP immunostaining data is accurate, the authors can only associate a change in APP processing with other toxic changes. There is no data to establish that the APP fragment detected is causative of any other characteristic of these cells. siRNA knockdown of APP would help establish causation.

It is not clear to me how the authors can conclude that this cell model is a valid model of AD. The data suggest that soon after differentiation, neurons with mutant PS1 would be induced to die, which clearly does not happen in these individuals.

7. PLOS authors have the option to publish the peer review history of their article (what does this mean?). If published, this will include your full peer review and any attached files.

Reviewer #1: No

Reviewer #2: No

---

## [Author Response · Author response to Decision Letter 1]

5 May 2020

We are grateful to editor and reviewers for the time spent and pertinent suggestions intended to improve our manuscript. The response (R/) to the specific comments is listed below. 

Dear Authors,

thank you for the submission and we truly appreciate the hard work in answering the previous concerns. There are some minor comments related to the result sections and data interpretation pertaining to aim APP and Abeta 42 levels reporting in the figures.

Please address these concerns.

Thank you

Response (R/). Minor comments have fully been addressed. Thank you. 

Reviewers' comments:

Reviewer's Responses to Questions

Comments to the Author

1. If the authors have adequately addressed your comments raised in a previous round of review and you feel that this manuscript is now acceptable for publication, you may indicate that here to bypass the “Comments to the Author” section, enter your conflict of interest statement in the “Confidential to Editor” section, and submit your "Accept" recommendation.

Reviewer #1: All comments have been addressed

Reviewer #2: (No Response)

2. Is the manuscript technically sound, and do the data support the conclusions?

Reviewer #1: Yes

Reviewer #2: Partly

3. Has the statistical analysis been performed appropriately and rigorously?

Reviewer #1: Yes

Reviewer #2: I Don't Know

 4. Have the authors made all data underlying the findings in their manuscript fully available?

Reviewer #1: Yes

Reviewer #2: Yes

 5. Is the manuscript presented in an intelligible fashion and written in standard English?

Reviewer #1: Yes

Reviewer #2: Yes

 6. Review Comments to the Author

Reviewer #1: This manuscript described a new and effective way of cell culture model that provides novel opportunities to study the mechanisms of AD on cellular level, which potentially will become a good model for therapeutic study. The data is very impressive. The corrected version already addressed all my comments and included a lot of extra effort. It is a good study.

Reviewer #2: The authors have largely addressed my original criticisms. However, the new data raise a new question that requires clarification. The authors indicate that the induced neurons with the PS1 mutation have increased levels of an APP fragment. In Figure 3, the authors have 2 time points post differentiation – 0 day and 4 days. I really see also most no change in the levels of most of what is measured over the 4 days. In Figure S3, the authors show that MSCs in regular medium with the PS1 mutation have no detectable levels of the intracellular APP fragments. I am struggling to understand why the 0 day for the mutant cells in the differentiation paradigm do not match the undifferentiated MSCs? A similar question applies to the oxidized DJ-1 data. 

R/ For the sake of clarity, the sentence “Western blot measurements and immunofluorescence analysis revealed that both WT and PSEN1 E280A MSCs displayed undetectable levels of intracellular APP / Aβ42 and oxidized DJ-1 at days 0, 2 and 4 post differentiation (S3A-J Fig)” has been changed to “After 7 days of culture in MCm, MSCs were left in RCm for 0, 2 and 4 additional days. Western blot measurements and immunofluorescence analysis revealed that both WT and PSEN1 E280A MSCs displayed undetectable levels of intracellular APP / Aβ42 and oxidized DJ-1 at days 0, 2 and 4 (S3A-J Fig)” (p24, L9-10). Briefly, MSCs were culture in non-differentiation media (i.e., in MCm) and differentiation media (i.e., Ch-N-Rm) for 7 days. After this time, both non-differentiated (i.e., MSCs) and differentiated cells (i.e., ChLNs) were exposed to RCm for 0, 2, and 4 additional days. 

The only proteins showing an obvious temporal change in levels are Caspase 3 and p53. In the Discussion, the authors state that day 0 is after 7 days of differentiation – in which case day is really day 7 and day 4 is really day 11. Please be precise in your descriptions.

R/ As rightly inferred by the referee, day 0 corresponded to 7 days of differentiation. For clarity, the sentence “Since Ch-N-Rm contains several factors that might interfere with the experiment interpretation and measurements, WT PSEN1 and PSEN1 E280A ChLNs (obtained after 7 days in Ch-N-Rm) were left in regular culture medium (RCm) for 0, 2 or 4 additional days, hereafter 0, 2, or 4 days of post transdifferentiation” is now added (p 10, L22-23; p11 L1-2). Please also note that, for most of the procedures, the sentence “ChLNs were left in regular medium (RCm) for 0, 2 and 4 days” was included in the original manuscript. 

The abstract mentions increased secretion of Abeta 42 by the mutant cells, but I did not see any data to support this conclusion.

R/ Please note that increased secretion of Abeta 42 by the mutant cells was described in the original version of this work in Materials and methods section (item Measurement of A�1-42 peptide in culture medium (p17) and data were shown in Fig. 6A.

If the APP immunostaining data is accurate, the authors can only associate a change in APP processing with other toxic changes. There is no data to establish that the APP fragment detected is causative of any other characteristic of these cells. siRNA knockdown of APP would help establish causation. 

R/ As mentioned in discussion section “Whether intracellular APP fragments build up��because full length APP processing is altered and consequently some APP fragments remain intracellular leading to cell death require further investigation. However, recent data support this view. Indeed, it has been shown that hAPP overexpression causes A�-independent neuronal death in olfactory sensory neurons via intrinsic apoptosis pathway [5, 6]” (p35, L2-6). Additionally, when comparing wild type versus mutant PSEN1 E280A ChLNs, all deleterious effects observed (e.g., OS, tau-phosphorylation, etc.) appeared only in mutant ChLNs simultaneously with appearance of sAPP�f, identified by LC-MS/MS. Taken together these observations suggest that sAPP�f is causative of herein described observations. We thank the reviewer for the suggestion to siRNA knockdown of APP. However, this procedure is out of scope of the present investigation, and will be explored in a near future. 

It is not clear to me how the authors can conclude that this cell model is a valid model of AD. The data suggest that soon after differentiation, neurons with mutant PS1 would be induced to die, which clearly does not happen in these individuals.

R/ For the sake of clarity, the sentence “…are a valid model of FAD and…” in abstract has been deleted. Effectively, further investigation in newborn brains bearing the PSEN1 E280A mutation is needed to fully establish whether or not this mutation accelerate neural cell death in these individuals. However, recent data suggest that “Studies from this cohort suggest that structural and functional brain abnormalities - such as cortical thinning and hyperactivation in memory networks - as well as differences in biofluid and in vivo measurements of Alzheimer's-related pathological proteins distinguish Presenilin1 E280A mutation carriers from non-carriers as early as childhood, or approximately three decades before the median age of onset of clinical symptoms” (in ref. [13] Fuller JT, et al. Biological and Cognitive Markers of Presenilin1 E280A Autosomal Dominant Alzheimer's Disease: A Comprehensive Review of the Colombian Kindred. J Prev Alzheimers Dis 2019;6(2):112-120). 

7. PLOS authors have the option to publish the peer review history of their article (what does this mean?). If published, this will include your full peer review and any attached files.

Do you want your identity to be public for this peer review? For information about this choice, including consent withdrawal, please see our Privacy Policy.

Reviewer #1: No

Reviewer #2: No

---

## [Decision Letter · Decision Letter 2]

7 May 2020

Cholinergic-like neurons carrying PSEN1 E280A mutation from familial Alzheimer’s disease reveal intraneuronal sAPPβ fragments accumulation, hyperphosphorylation of TAU, oxidative stress, apoptosis and Ca2+ dysregulation: Therapeutic Implications

PONE-D-19-22448R2

Dear Dr. Jimenez-Del-Rio,

We are pleased to inform you that your manuscript has been judged scientifically suitable for publication and will be formally accepted for publication once it complies with all outstanding technical requirements.

With kind regards,

Firas H Kobeissy, PhD

Academic Editor

PLOS ONE

Additional Editor Comments (optional):

Reviewers' comments:

Reviewer's Responses to Questions

**Comments to the Author**

1. If the authors have adequately addressed your comments raised in a previous round of review and you feel that this manuscript is now acceptable for publication, you may indicate that here to bypass the “Comments to the Author” section, enter your conflict of interest statement in the “Confidential to Editor” section, and submit your "Accept" recommendation.

Reviewer #2: All comments have been addressed

2. Is the manuscript technically sound, and do the data support the conclusions?

Reviewer #2: (No Response)

3. Has the statistical analysis been performed appropriately and rigorously? 

Reviewer #2: I Don't Know

4. Have the authors made all data underlying the findings in their manuscript fully available?

Reviewer #2: Yes

5. Is the manuscript presented in an intelligible fashion and written in standard English?

Reviewer #2: Yes

6. Review Comments to the Author

Reviewer #2: The authors have addressed all my concerns. Sorry I missed the figure on Abeta levels in the revision. I think I was expecting the data to be after Fig 3.

7. PLOS authors have the option to publish the peer review history of their article (what does this mean?). If published, this will include your full peer review and any attached files.

Reviewer #2: No

---

## [Editor Report · Acceptance letter]

8 May 2020

PONE-D-19-22448R2 

Cholinergic-like neurons carrying PSEN1 E280A mutation from familial Alzheimer’s disease reveal intraneuronal sAPPb fragments accumulation, hyperphosphorylation of TAU, oxidative stress, apoptosis and Ca2+ dysregulation: Therapeutic Implications 

Dear Dr. Jimenez-Del-Rio:

I am pleased to inform you that your manuscript has been deemed suitable for publication in PLOS ONE. Congratulations! Your manuscript is now with our production department. 

With kind regards,

on behalf of

Dr. Firas H Kobeissy 

Academic Editor

PLOS ONE